# A microfluidic device for inferring metabolic landscapes in yeast monolayer colonies

Zoran S Marinkovic[1,2,3], Clément Vulin[1,4,5], Mislav Acman[1,3], Xiaohu Song[2], Jean-Marc Di Meglio[1], Ariel B Lindner[2,3]*, Pascal Hersen[1]*

[1]Laboratoire Matière et Systèmes Complexes, UMR 7057 CNRS and Université de Paris, Paris, France; [2]U1001 INSERM, Paris, France; [3]CRI, Université de Paris, Paris, France; [4]Institute of Biogeochemistry and Pollutant Dynamics, ETH Zürich, Zürich, Switzerland; [5]Department of Environmental Microbiology, Eawag, Dübendorf, Switzerland

**Abstract** Microbial colonies are fascinating structures in which growth and internal organization reflect complex morphogenetic processes. Here, we generated a microfluidics device with arrays of long monolayer yeast colonies to further global understanding of how intercellular metabolic interactions affect the internal structure of colonies within defined boundary conditions. We observed the emergence of stable glucose gradients using fluorescently labeled hexose transporters and quantified the spatial correlations with intra-colony growth rates and expression of other genes regulated by glucose availability. These landscapes depended on the external glucose concentration as well as secondary gradients, for example amino acid availability. This work demonstrates the regulatory genetic networks governing cellular physiological adaptation are the key to internal structuration of cellular assemblies. This approach could be used in the future to decipher the interplay between long-range metabolic interactions, cellular development and morphogenesis in more complex systems.
DOI: https://doi.org/10.7554/eLife.47951.001

*For correspondence:
ariel.lindner@inserm.fr (ABL);
pascal.hersen@univ-paris-diderot.fr (PH)

**Competing interests:** The authors declare that no competing interests exist.

## Introduction

Structured cellular populations are complex, dynamic systems and their composition, expansion and internal structure are the result of interactions between the cells and their microenvironment. Cells absorb and metabolize nutrients and also produce and secrete metabolites, creating spatial gradients of nutrients and metabolites. Thus, cells at the outskirts of a multicellular assembly do not experience the same microenvironment as the cells deeply buried within. Reciprocally, cellular physiology is dependent on the cell's position within a colony. Such variations in cellular physiology are consistently observed in a variety of multicellular systems – from bacterial and yeast colonies (*Vulin et al., 2014*; *Cáp et al., 2012*) to biofilms (*Nadell et al., 2016*) and tumors (*Carmona-Fontaine et al., 2013*; *Delarue et al., 2014*) – and are reflected by altered gene expression levels and cellular phenotypes as growth rates, nutrient uptake rates and metabolic activity. Such variations presumably emerge because of long-range metabolic interactions between cells, in that the cellular microenvironment at one position depends on the nutrient uptake rate at another position.

Notably, multicellular communities (*Shapiro, 1998*; *Shou et al., 2007*; *Xavier and Foster, 2007*) exhibit various adaptive benefits, including higher cell proliferation, improved access to resources and niches (*Koschwanez et al., 2011*), collective defence (*e.g.*, against antagonists, drugs, antibiotics) (*Nadell et al., 2016*) resulting in optimization of population survival when confronted with averse physical, chemical, nutritional or biological challenges (*Palková and Váchová, 2006*). These

examples indicate the importance of understanding the emergence and maintenance of complex spatial multicellular structures from ecological (*Antwis et al., 2017*; *Gonzalez et al., 2012*; *Widder et al., 2016*), medical (*Bryers, 2008*; *Gilbert et al., 2018*; *Estrela and Brown, 2018*) and evolutionary (*Ratcliff et al., 2012*; *Nadell et al., 2010*; *Kim et al., 2008*) perspectives. Yet, despite the obvious contrast between homogeneous environments and the pronounced environmental heterogeneity of microbial cellular assemblies, the majority of scientific research to date has either focused on single cells in homogeneous environments or populations of cells grown in batch or continuous liquid cultures. This is mostly due to the complexity of designing an experiment that would allow monitoring, over long time, the development of a spatially defined extended multicellular assembly. This is in particular the case for the widely used eukaryotic model organism yeast *Saccharomyces cerevisiae*, despite the numerous calls in recent reviews to study its nutrient sensing, signaling, and related growth and development control within the natural colony context. (*Conrad et al., 2014*; *Broach, 2012*; *Horák, 2013*).

As microorganisms in nature tend to live in multicellular communities, devising an experimental approach that captures this complexity while being easy to use and amenable to different experimental needs and conditions should further our understanding of complex gene regulatory networks in the context of microbial evolution and ecology.

Current direct observations of three-dimensional colonies and biofilms are cumbersome and often constrained by existing technologies (*Nadell et al., 2016*). For example, two-photon microscopy of sliced agarose-encapsulated yeast colonies was required to show that yeast cells may adopt different physiologies – and possibly different cell types – depending on their position within a colony (*Cáp et al., 2012*). In another example, nanospray desorption electrospray ionization mass spectrometry (nanoDESI MS) was used to study growing bacterial colonies on agar plates and showed a wide diversity and complexity of compounds that characterize microbial chemical ecology (*Watrous et al., 2012*; *Traxler and Kolter, 2012*). Such complex methodologies are not amenable to time-lapse imaging, nor to observation of the temporal variations in gene expression and growth rates of single cells over relevant time and length scales. An alternative is to grow microbial cells in microfluidic devices to spatially constrain the growth of the cells and to control the delivery of nutrients (*Bennett and Hasty, 2009*; *Cookson et al., 2005*; *Robert et al., 2010*; *Ni et al., 2012*; *Llamosi et al., 2016*). Microfluidic experimental research is typically designed to ensure that the cells being studied experience a homogeneous environment. This can be done at the single cell level, as it has been demonstrated in studies of aging of yeast (*Jo et al., 2015*) and bacteria (*Yang et al., 2019*) where single cells had to be trapped and kept under constant nutrient flow for long term observations to capture their death. Alternatively, a small cell assembly can be trapped in dead-end chambers under assumption that a quick diffusion of nutrients will keep the environment in chambers homogeneous. With that approach cell lineages were tracked for bacteria (*Wang et al., 2010*) (the widely used 'mother machine') and yeast (*Xu et al., 2015*), cells were subjected to fluctuating environments of different carbon sources to study non-genetic memory in bacteria (*Lambert et al., 2014*), and bacterial colonies were synchronized through quorum sensing and gas-phase redox signalling over centimeter-length scales to produce oscillating colony 'biopixels' (*Prindle et al., 2011*). Although effective for their specific applications, unfortunately, such devices do not fully capture emerging properties at a colony level, that is spatial variations in growth rates, microenvironments and phenotypes. Recently, there have been a few attempts to use microfluidics to study collective properties of bacterial colonies grown as a microcolony. *Hornung et al. (2018)* grew two-dimensional bacterial microcolonies in a 75 μm long device perfused with a very low concentration (up to 585 μM) of protocatechuic acid as the only carbon source from both sides of the cell chamber. They observed heterogeneous growth in agreement with a combination of a reaction-diffusion model and particle-based simulations. In a similar setup, a 60 μm long device perfused with a very low concentration of glucose (up to 800 μM) from one side was used to study the emergence of microscale gradients that resulted in metabolic cross-feeding between glucose-fermenting and acetate-respiring subpopulations of bacteria and antibiotic tolerance by slow growing subpopulation (*Co et al., 2019*). Wilmoth et al. used microwells up to 100 μm in diameter to look at spatial patterns of H1-Type VI secretion system (T6SS) mutants of *Pseudomonas aeruginosa* accompanied with an agent-based model depicting the two observed subpopulations (*Wilmoth et al., 2018*). They found that spatial constraints and local concentrations of growth substrates affect the spatial organization of cells. Finally, Liu et al. grew *Bacillus subtilis* biofilms perfused with glycerol and glutamate media.

They discovered collective oscillations which emerge as a consequence of long-range metabolic co-dependence between cells in the interior and cells at the periphery of a biofilm, presumably to maximize the availability of nutrients and survival of interior cells (*Liu et al., 2015*). While the use of microfluidics gave rise to the discovery of collective properties of microbial assemblies and facilitated quantitative observations, such attempts are currently limited by inherent difficulties tied to fabrication of efficient microfluidic devices and choice of model organism. These limitations include small device dimensions (<100 µm) (*Hornung et al., 2018*; *Co et al., 2019*; *Wilmoth et al., 2018*), use of low nutrient concentrations (<1 mM) (*Hornung et al., 2018*; *Co et al., 2019*), limited scope of nutrient types (*Hornung et al., 2018*; *Co et al., 2019*; *Liu et al., 2015*) and in some cases inability to access single cell level measurements (*Wilmoth et al., 2018*). Therefore, it is challenging to apply such devices for the general case of the study of a large monolayer of cells in standard range and scope of nutrients, often used in biological research in liquid cultures. Additionally, it is tempting to reconstruct the emergence of gene expression landscapes on a global scale (*e.g.,* within structured populations) from local (*e.g.,* single cell) properties, given the extensive knowledge accumulated on single-cell gene regulatory networks. However, the variations in the microenvironment within a multicellular assembly and their interconnections with gene expression and cell metabolism are rather poorly known.

In attempt to overcome the above limitations in current methodologies and observe emerging properties at a colony level in larger dimensions and standard nutrient conditions, we developed a microfluidic device to grow thin, extended arrays of yeast cell monolayers that are perfused with nutrients from a single direction. We demonstrate the emergence of heterogeneous microenvironments and quantify spatial variation in cellular growth rate and the formation of gene expression landscapes for key metabolic genes involved in glucose transport and utilization, across the nascent 2D microcolony in 800 µm long cell chambers and up to 444 mM glucose concentration. Interestingly, the gene expression landscapes exhibited a high degree of spatial correlation over a wide range of glucose concentrations. Notably, we show that a growing extended assembly of cells presents a robust, steady state spatial structure, transitioning between fermentative (high glucose environment, fast growth, rapid glucose utilization) and respiratory (low glucose environment, slow growth, slow but efficient glucose utilization [*Pfeiffer and Morley, 2014*; *Hagman and Piškur, 2015*]) regimes, located close to and far from the nutrient source, respectively. This spatial structure emerges from the interplay between how cells individually adapt to the microenvironment and, at the same time, alter their surroundings as a result of their metabolic activity. Said differently, structured cells create and experience a spatially structured micro-environment through the interplay of nutrient diffusion and uptake without any obvious inherent biological program that would imply cell-cell communication and coordinated community action.

## Results

### Growing extended yeast monolayers

Microfluidic systems are usually designed to ensure a homogeneous microenvironment for all cells (*Bennett and Hasty, 2009*). In contrast, in this study, we designed a microfluidic device – dubbed the 'yeast machine' – to grow long, narrow yeast monolayers with the aim of observing the emergence of nutrient gradients and spatial variations in cellular growth and gene expression landscapes. We used soft lithography techniques to fabricate a multi-layered microfluidic device composed of a large channel (to flow nutrients) and an array of perpendicular, extended (800 µm-long), narrow (50 µm-wide), flat (4.5 µm-high) dead-end chambers in which yeast cells can grow as monolayers while the media is supplied by a pressure pump-based system with flow control (*Figure 1*, *Figure 1—figure supplement 1*). The length of the dead-end chambers was optimized to induce significant variations in the nutrient concentrations within the chambers due to cellular nutrient uptake. The chamber width was large enough to avoid jamming during cell growth due to geometric constraints and small enough to avoid generation of complex, cell-recirculating flows induced by cell growth (*Boyer et al., 2011*). The chamber height was comparable to – but slightly larger than – the average size of a yeast cell, so the cells were vertically constrained to facilitate single-cell imaging and time-lapse fluorescence microscopy (*Figure 1—figure supplement 2*).

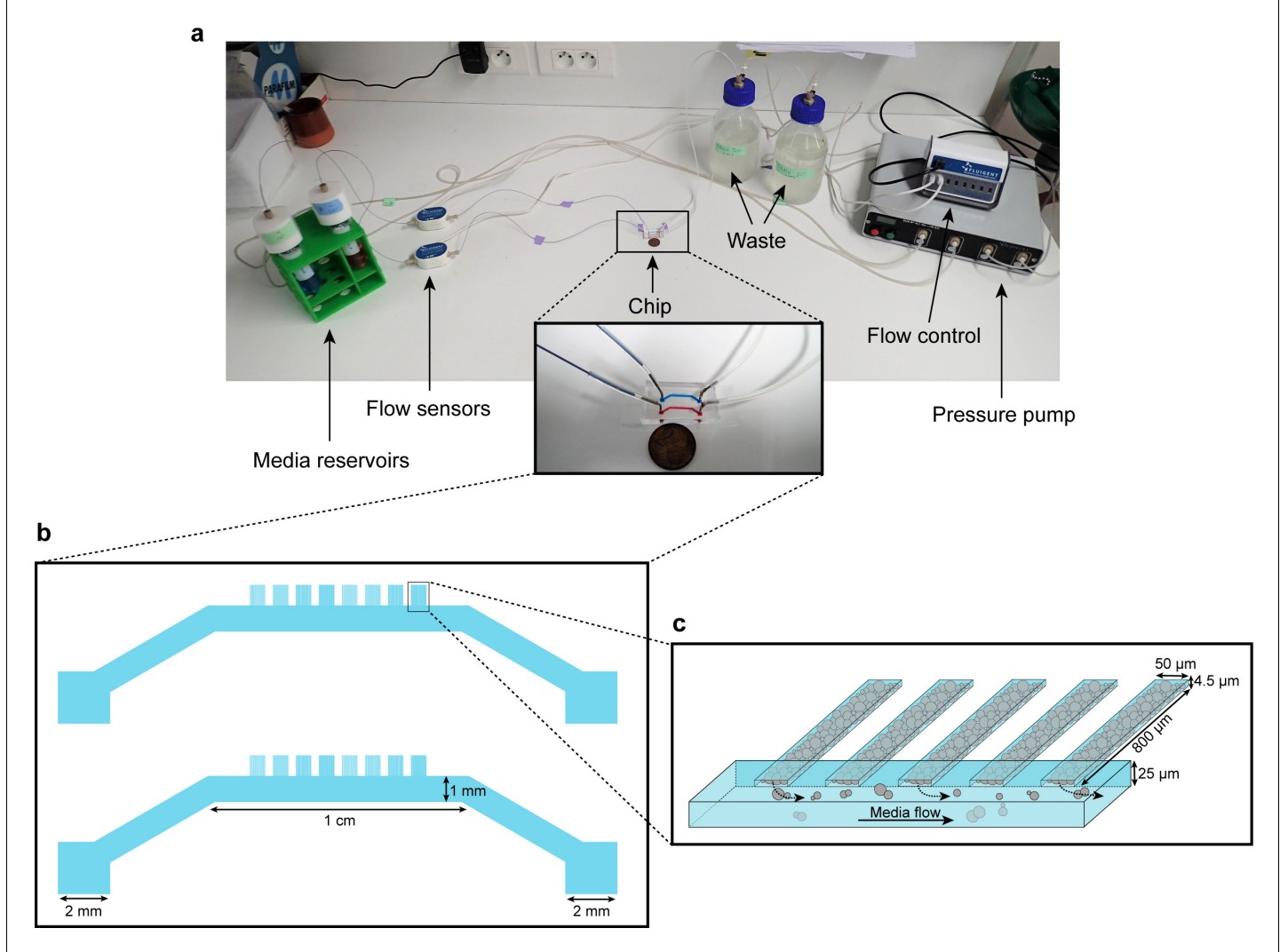

**Figure 1.** Microfluidic device setup and design.  (a) Media reservoirs are pressurized with the help of Fluigent MFCS pressure pump resulting in flow through the flow sensors, into the chip and then to waste. Flow sensors and pressure pump are connected to the flow-rate control module, which maintains a constant flow through the system. Nutrient supply and media conditions can be changed in real time. (b) Each single 'yeast machine' has two sets of cell chambers of various widths (5 µm, 10 µm, 25 µm and 50 µm). The Cell chambers are connected perpendicularly to a large flow channel (1 mm wide, 25 µm high). This design facilitates adaptation for different model systems (*e.g.* bacteria, yeast, mammalian cells) and high-throughput depending on the of predefined flexible length, width and height flexible adapted dimensions. (c) A close-up sketch of a set of cell chambers used in our experiments. They are 800 µm long, 50 µm wide, and 4.5 µm high. A single cell chamber fits a monolayer of up to 2500 yeast cells. The whole setup is mounted on a microscope for time-lapse fluorescent imaging.

DOI: https://doi.org/10.7554/eLife.47951.002
The following figure supplements are available for figure 1:

**Figure supplement 1.** Experimental Details.
DOI: https://doi.org/10.7554/eLife.47951.003
**Figure supplement 2.** Detailed cell chamber view.
DOI: https://doi.org/10.7554/eLife.47951.004

The cells were injected into the main channel of the 'yeast machine' and then forced into the dead-end chambers by centrifugation using a homemade 3D-printed holding device attached to a spin coater (see *Figure 1—figure supplement 1*; Materials and methods). The main channel was washed with yeast synthetic complete growth medium to remove excess cells; cells that were trapped in the dead-end chambers by centrifugation were not removed by the washing step. Nutrients were flowed through the main channel and could passively diffuse into the array of dead-

end chambers. The cells formed growing monolayers that extended from the closed end of the chamber and collectively progressed towards the nutrient source (*i.e.* the open end of the chamber) as the cells pushed each other while growing (*Figure 2a,b*; *Video 1*). Cells eventually filled each chamber, forming an extended two-dimensional colony composed of about 2500 cells (*Figure 2b*, *Figure 1—figure supplement 2*), typically ~10 cells wide and ~200 cells long. Cells could be observed locally at high magnification (100 × objective), while the whole assembly could be seen at low magnification (10 × objective). We recorded the cellular expansion and subsequent internal dynamics of these long monolayers, as well as the landscape of expression of key fluorescently tagged endogenous genes, over time and over an almost 1000-fold range of glucose concentrations (from 0.01% to 8% *w/vol*).

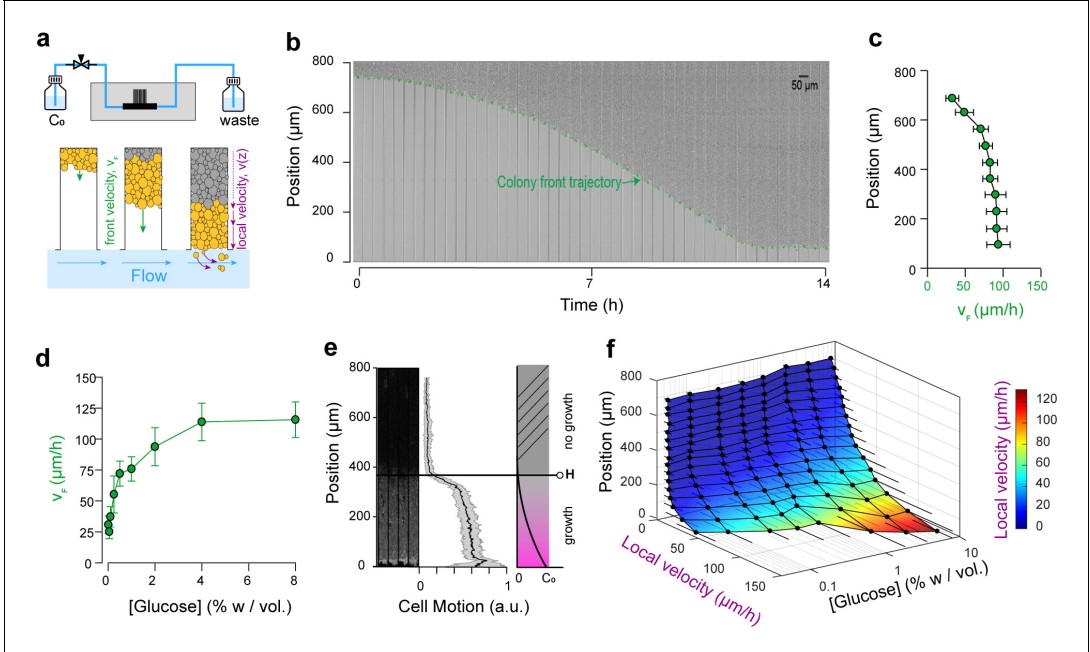

**Figure 2.** Expansion and dynamics of extended cellular monolayers. (a) The microfluidic device is perfused with nutrients using a pressure-driven system (see *Figure 1*). Yeast monolayers extend within the long chambers: front velocity ($V_F$) and local velocity ($Vz$) are determined by cellular growth and division. (b) Example of a time-lapse collage of yeast monolayer expansion along an 800 µm-long chamber (2% *w/vol*. glucose, 5 × amino acid concentration). Front velocity increases and reaches a plateau (indicated by flattening of the slope of the green curve). When the front approaches close to the open end of the chamber (*i.e.*, 0 µm), the over-spilling cells are constantly washed away by the nutrient flow within the main channel. (c) Front velocity reaches a maximum when the position of the front becomes close to the open end of the chamber indicating that after expanding by a typical distance (~400 µm here for 2% *w/vol*. glucose), the maximal number of cells that receive glucose and can participate in expansion has been reached. 340 velocity data points binned into 10 equally spaced position points were extracted from n = 12 colony front trajectories (2% *w/vol*. glucose). The error bars denote standard deviations of each bin (~15–30 velocity data points). (d) Front velocity as function of external glucose concentration. Data comes from the bin closest to the open end of the chamber as measured in *Figure 2c* for each glucose concentration (n > 5). Error bars denote standard deviations. (e) Local cellular motion can be assessed by computing the standard deviation of pixel intensities across a stack of time-lapse images. Here, white areas indicate variations in movement across the time-lapse for cells below 400 µm, while the cells above do not move. Averaging over several channels (n = 9), we obtained an indicator of cell motion and thus an estimate of the glucose penetration distance, $H$ (~400 µm for 2% glucose). (f) Local velocity decreases for cells deeper within the chamber. Local velocity also increases with external glucose concentration. Velocity Data, that were binned into 16 equally spaced position, comes from the analysis of >100 cell trajectories. Error bars denote standard deviations.

DOI: https://doi.org/10.7554/eLife.47951.006

The following figure supplements are available for figure 2:

**Figure supplement 1.** Comparison of front velocity and local velocity under low and high amino acid concentrations.
DOI: https://doi.org/10.7554/eLife.47951.007

**Figure supplement 2.** Local velocity and front velocity over a range of external glucose concentrations, $C_0$.
DOI: https://doi.org/10.7554/eLife.47951.008

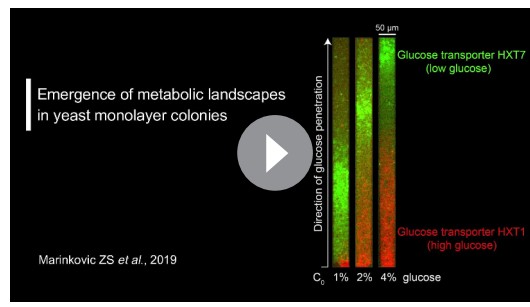

**Video 1.** Emergence of metabolic landscapes in yeast monolayer colonies.
DOI: https://doi.org/10.7554/eLife.47951.005

## Monolayer in expansion displays regions of fast and slow growth

Expansion of the monolayers of cells was observed by microscopy at low magnification (10× objective). Under standard glucose-rich conditions (2% *w/vol*; 111 mM) and excess amino acids (5× CSM, see Materials and methods), the front velocity, $V_F$, increased during the first 2-4 h and eventually reached a steady-state close to 100 µm.h$^{-1}$ (*Figure 2c,d*, *Video 1*). Front velocity is the sum of the contribution of every cell to colony expansion. Therefore, $V_F$ depends on the quantities of glucose and other nutrients that penetrate inside the yeast monolayer, which impact both the number of cells that grow and their growth rates. Initially, the monolayer is sparsely populated and sufficient glucose is expected to reach all cells. After growth and division, a larger number of cells can participate in global expansion of the population. Thus, the front velocity is expected to quickly increase over time. However, at some point, as the size of the monolayer increases, the cells close to the dead end of the chamber will stop growing (due to absorption and metabolism of available nutrients by cells closer to the nutrient source/chamber opening) and the front velocity will plateau. Hence, after the cell chamber populates with cells completely, a steady-state is reached where a constant number of cells with access to glucose continue to divide and move passively towards the nutrient source, while the number of cells at the dead end of the chamber deprived of glucose (and other nutrients) remains unchanged. If we consider the ideal case in which yeast cells are 4 µm-wide and divide every 90 min in the presence of glucose, each cell layer leads to an expansion of 4 µm every 90 min, or 2.6 µm.hr$^{-1}$. The observed terminal front velocity of 94 ± 8 µm.hr$^{-1}$ (*Figure 2*) can be attributed to the first 36 ± 3 layers of cells, that is the first 140 µm of the colony. The glucose penetration distance can be approximated by assuming (*Vulin et al., 2014*) that glucose – of which the concentration is maintained at $C_0$ at the front of the monolayer – freely diffuses within the assembly with a diffusion coefficient $D \sim 100$ µm (*Cáp et al., 2012*).s$^{-1}$ and is absorbed by cells at a constant rate, $q_0$, of ~ 1 mM.s$^{-1}$. Diffusion law dictates that the glucose concentration is expected to decrease significantly after a typical distance, $H$, that scales with $\sqrt{\frac{DC_0}{q_0}} \sim 100$ µm. Our direct observation (*Figure 2e*) showed that for a layer of growing cells, $H$ is around 400 µm at 2% *w/vol* glucose. Notably, both estimations are in agreement, albeit they underestimate the observed size of the growing layer. These discrepancies result from discarding the decay in the cellular growth rate at decreasing glucose concentrations and the variation in the specific cellular uptake rate, $q$, with glucose concentration. Indeed, the interplay between glucose diffusion and uptake is central to structuration of the colony as it affects both the number of cells that have access to glucose and the glucose concentration in the microenvironment of each region, and thus determines which cells actually participate in colony expansion and by how much (*Vulin et al., 2014*). The true glucose penetration distance is therefore likely to be larger than the above 'guesstimate'. Yet, inferring the true penetration distance would require a detailed model of the dependency of both cellular glucose absorption and the growth rate on the glucose concentration, as well as experimental measurements of the glucose concentrations within the monolayer. This outlines the difficulty of predicting the internal structure of a simple yeast monolayer due to our limited understanding of how yeast cells interact with nutrients and the difficulty of obtaining quantitative details of the microenvironment landscapes within a yeast monolayer. In the following text, we quantify the expression of different glucose concentration-dependent transporters as a possible proxy for intra-colony glucose concentration. We even ventured further, to study how landscapes of cellular growth and expression of key genes involved in glucose transport self-emerge from long-range metabolic interactions within the yeast colony.

### Front velocity increases with glucose concentration

Increasing the glucose concentration (from 0.01% to 8% *w/vol*) led to higher terminal front velocities (*Figure 2d*), in agreement with the fact that at higher concentrations, glucose will penetrate further

by diffusion in the colony (*Figure 2a*). Thus, increasing the concentration allows a larger number of cells to access glucose and participate in the growth of the colony. Yet, the front velocity does not increase linearly with glucose concentration, and plateaus at very high glucose concentrations (>4% *w/vol*). One interpretation is that at this concentration range, sufficient glucose reaches the dead end of the chamber, allowing all cells to participate in the growth of the assembly. However, based on $V_F \sim \mu L$, where $L$ is the length of the dead-end chamber and $\mu$ is the average cell growth rate, one would expect a saturating front velocity of 368 $\mu m.h^{-1}$, much larger than the measured value of 100 $\mu m.h^{-1}$.

Glucose is not the only nutrient required for cellular growth; amino acids can be a limiting factor for auxotrophic strains such as the one employed in this study (S288C background). This is why we used an excess of amino acids (5 × CSM) compared to classic SC medium for yeast cell cultures. Indeed, using standard amino acid concentrations in the media resulted in significantly lower terminal front velocities, even at high glucose concentrations (*Figure 2—figure supplement 1*). This suggests that amino acid availability can limit cellular growth, which is especially visible in the presence of high glucose concentrations, where glucose is no longer limiting but amino acids are. As all experiments were performed under 5-fold higher amino acid concentrations than normal SC medium, other metabolites that are consumed are likely to form gradients within colony and might become rate-limiting for growth. Taken together, we conclude that the spatial variations in all metabolic components of the microenvironment need to be taken into account in order to fully understand microbial colony growth. With that in mind, building a mathematical model to account for the observed expansion of a spatially structured colony is barely achievable, and we will not address this question here. Rather, we opted to further characterize the development of glucose gradients as a specific and critical component of the emergence of the metabolic landscape of the colony.

## Local expansion rate decreases with distance from the nutrient source

Once the dead-end chambers were filled with cells, we found that similar growth pattern emerged across parallel chambers, specific to each glucose concentration. The cells closer to the open end of the chambers continued to divide, pushing cells out that were washed away by the flow in the nutrient channel. Cells closer to the dead end ($y \sim 800$ $\mu m$) did not move, grow nor divide. At standard glucose conditions (2% *w/vol*) and a high amino acid concentration (5 × CSM), significant cell motion was not observed after $y \sim 400$ $\mu m$, indicating that very limited glucose is available to the cells that are beyond this region. By tracking single cell trajectories, we measured the velocity field within the yeast monolayers over a range of glucose concentrations. We extracted >100 single cell trajectories per concentration, resulting in thousands of velocity data points (see Materials and methods). As expected, increasing the glucose concentration in the nutrient channel (from 0.01% to 8% *w/vol*) led to higher local velocities deeper in the colony (*Figure 2f*, *Figure 2—figure supplement 2*). Concomitantly, velocity also increased closer to the chamber opening when cells experienced a higher glucose concentration.

In summary, our setup captures the essence of structured colonies, with the emergence of a landscape of growth divided into a non-growing area and actively growing area. This spatial separation is the result of the formation of glucose (and other nutrient) gradients. These gradients emerge as a result of cellular metabolic activity, which in turn affects the cellular growth rate and physiology at the local scale.

## Cellular metabolic activity creates gene expression landscapes

The emerging glucose (and other nutrient) gradients are expected to both trigger and be governed by differential gene expression landscapes. To this end, we studied the expression of seven key glucose transporters (HXT1-7) whose expressions are regulated by the extracellular glucose concentration. We employed yeast strains in which these endogenous glucose transporters were tagged with GFP (Materials and methods), and recorded the fluorescence signals at the global scale using a low-magnification objective (10×) and local cellular scale using a high-magnification objective (100×). Cells were loaded into the chambers as described above and observed after the establishment of a quasi-steady state (starting 10 hr after the chamber was filled with cells, *Figure 3—figure supplement 1*). We observed the formation of different landscapes of gene expression for each of the seven transporters, each with marked territories of low and high expression (*Figures 3* and

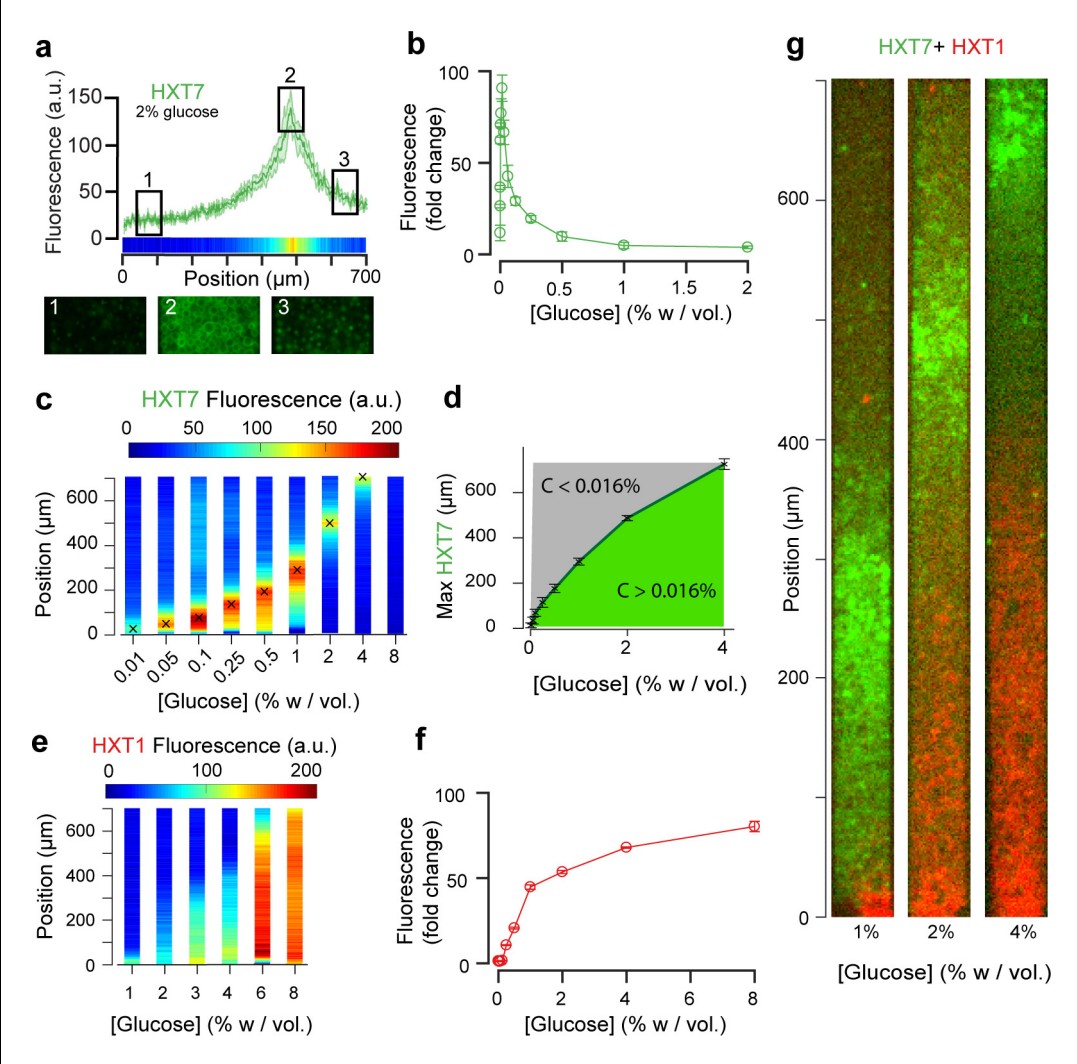

**Figure 3.** Landscapes of gene expression self-emerge in extended yeast monolayers. (**a**) Expression profile of HXT7-GFP along the chamber (average fluorescence levels, n = 9; standard deviation shown as the envelope) for an external concentration of 2% *w/vol* glucose. Membrane localization of HXT7 was only observed in the cells surrounding the area of peak HXT7 expression, localized at ~500 μm at 2% *w/vol.* glucose. (**b**) FACS measurements of HXT7-GFP expression in batch culture (average of three replicates) showing a single intensity peak at $C_0$ = 0.016%. This peak value can be mapped back to the spatial landscape of 3a to infer the glucose concentration in the region of peak HXT7-GFP fluorescence. n = 3–6 per glucose concentration (**c**) On varying the glucose concentration in the nutrient channel, we observed a transition in peak HXT7-GFP fluorescence within the 2D colony. At a concentration of 4% *w/vol* and above, the peak was located close to the dead end of the chamber or not visible, indicating sufficient glucose was available throughout the chamber (color code normalized to maximal expression level). Data obtained from n = 8–17 replicates per glucose concentrations (see also *Figure 3—figure supplement 2*). (**d**) Compared with 3b, it is possible to roughly define areas of glucose presence in the monolayer for a range of glucose concentrations (n = 8–17, per glucose concentrations, error bars denote ± one standard deviation). (**e**) Landscape of HXT1-GFP gene expression over a range of glucose concentrations (color code normalized to maximal expression); n = 8–9 per glucose concentrations (*Figure 3—figure supplement 3*). (**f**) FACS measurements of HXT1-GFP over a range of glucose concentrations; n = 3 replicates. **3** g. Overlay of HXT1 (red) and HXT7 (green) gene expression landscapes at three external glucose concentrations, showing that the expression landscapes of these transporters were inversely correlated, in agreement with their different glucose-dependent expression patterns (compare 3b and 3 f).
DOI: https://doi.org/10.7554/eLife.47951.009

The following figure supplements are available for figure 3:

**Figure supplement 1.** HXT1 and HXT7 landscape dynamics.

*Figure 3 continued on next page*

*Figure 3 continued*

DOI: https://doi.org/10.7554/eLife.47951.010

**Figure supplement 2.** Extended figure of *Figure 3c*.

DOI: https://doi.org/10.7554/eLife.47951.011

**Figure supplement 3.** Extended figure of *Figure 3e*.

DOI: https://doi.org/10.7554/eLife.47951.012

*4*; Materials and methods). In particular, HXT1 and HXT7 displayed inversely correlated landscapes of gene expression (*e.g.*, *Figure 3a and g* for 2% *w/vol* glucose). Both patterns demonstrate the formation and maintenance of a glucose gradient that emerges from cellular metabolic activity. HXT1 is a low-affinity glucose transporter mainly expressed under high-glucose conditions, while HXT7 is a high-affinity glucose transporter expressed under low-glucose conditions only (*Figure 3b and f*) (*Reifenberger et al., 1997*; *Diderich et al., 1999*; *Maier et al., 2002*). Concomitantly, HXT1 was expressed at the highest levels in the cells close to the chamber opening (*i.e.*, in the highest glucose concentration), while HXT7 expression peaked further away in the chamber, indicating a transition to a low-glucose region. We examined the cells at higher magnification ($60\times$) to assess the localisation of HXT7 gene expression. As expected, in the cells expressing the highest levels of this gene, the fluorescence was localized to the cell membrane, indicating HTX7 played an active role in glucose transport in these cells. In contrast, deeper in the colony, we observed lower levels of HTX7 fluorescence due to the long lifetime of GFP-fused proteins and absence of dilution through cell division, though this fluorescence was localized in vacuoles, indicating the transporter had been targeted for degradation by the cells (*Hovsepian et al., 2017*) (*Figure 3a*). Assuming the observed peak of HXT7 fluorescence matches the peak fluorescence observed in batch culture at a glucose concentration of 0.016% *w/vol.* (*Figure 3b,c*, *Figure 3—figure supplement 2*), we could locate the position in the yeast monolayer at which the glucose concentration reached 0.016% *w/vol*. This position was around $H_f \sim 500$ μm from the front, in good agreement with the transition in cell motion (*Figure 2*, $H_m \sim 400$ μm).

## Gene expression landscapes depend on the glucose source concentration

Increasing the glucose concentration in the nutrient channel changed the gene expression landscape of all seven glucose transporters (*Figures 3* and *4*). In particular, at 1% *w/vol* glucose, HXT1 was only expressed at low levels at the growing front of the colony ($y < 60$ μm). In contrast, at the highest glucose concentration (8% *w/vol*; *Figure 3e*, *Figure 3—figure supplement 3*), HXT1 was expressed at high levels throughout the whole colony, demonstrating glucose was available throughout the chamber. As HXT1 is mainly expressed under high-glucose conditions (>1% *w/vol* glucose) in batch culture (*Diderich et al., 1999*), this observation indicated the glucose penetration distance (within the chamber) increased with the external glucose concentration. This is in agreement with the increase in local velocity with the external glucose concentration in *Figure 2*, with the size of the growing area also increasing with the external glucose concentration.

In contrast, HXT7 exhibited a peak-like expression pattern, and was repressed under both high-glucose conditions and when no glucose was present. At low-glucose concentrations (0.1% *w/vol*), a peak in HXT7 expression was observed at the very beginning of the colony ($y \sim 20$ μm), indicating glucose was quickly absorbed by the cells closest to the chamber opening, thus these were the only cells with access to sufficient carbon resources to grow and divide. The peak of HXT7 expression moved deeper into the colony as the glucose concentration increased and disappeared completely at 8% *w/vol* glucose, again indicating sufficient glucose could diffuse to the end of the chamber under high-glucose conditions (*Figures 3* and *4*).

## Reconstructing glucose concentration landscapes using glucose transporter gene expression levels

We assessed the expression profiles of HXT1-7 in batch culture as a function of glucose concentration (see Materials and methods) to obtain a qualitative idea of the glucose concentrations within the microfluidic device. The data for HXT7 was particularly revealing: its rather sharp, well-defined

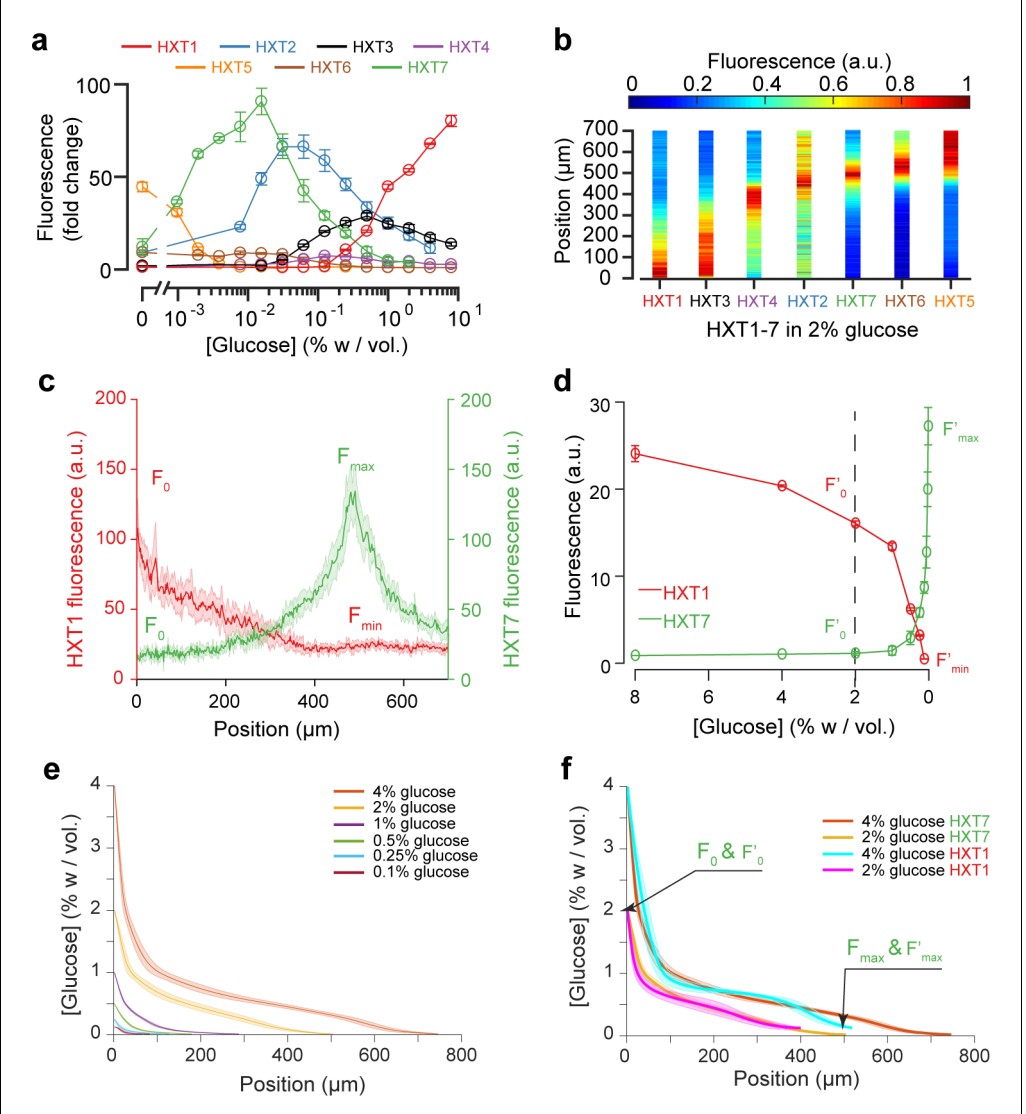

**Figure 4.** Using the fluorescence landscapes of glucose transporter gene expression to infer glucose concentration gradients. (**a**) FACS measurements for HXT1-GFP to HXT7-GFP in batch culture over a range of glucose concentrations. The expression levels of each HXT show a specific dependence on glucose concentration (n = 3–6 replicates per glucose concentration). (**b**) Landscapes of gene expression for all HXTs-GFP at an external glucose concentration of 2% *w/vol*. HXTs are ordered by their relative glucose specificity: HXT1 is expressed under high-glucose conditions, while HXT5 is only expressed at very low-glucose conditions. Assuming a progressive spatial decay in the glucose concentration away from the chamber opening, all maps of gene expression are in perfect agreement with the intensity profiles observed in batch culture (n = 8–10 replicates per glucose concentration). (**c-d**) Method of glucose gradient reconstruction. The fluorescence landscape of HXT7 (*resp.* HXT1) shows a peak $F_{max}$ (*resp.* a minimum, $F_{min}$) at a given location. The fluorescence intensity at the opening of the chamber, $F_0$, corresponds to the external glucose concentration, $C_0$. Using the FACS measurements of HXT7 (*resp.* HXT1) as a function of glucose concentration, one can define the concentration of glucose that matches the peak $F_{max}$ (respective to the minimum $F_{min}$), and the fluorescence intensity that corresponds to $C_0$. This allows us to linearly map all other fluorescence intensities for a given glucose concentration from the batch culture to the fluorescence intensities inside the colony, allowing the glucose concentration across the entire cellular monolayer to be reconstructed. Data comes from previously mentioned HXT1 and HXT7 microfluidics and flow cytometry measurements. (**e-f**) Reconstruction of glucose concentration obtained from HXT7 (**e, f**) and HXT1 (**f**) fluorescence data and various external glucose concentrations.

DOI: https://doi.org/10.7554/eLife.47951.013

expression peak at 0.016% *w/vol* allowed to define the distance in the microfluidic device at which the glucose concentration is close to that value (*Figure 3a,c*). This concentration boundary separates the yeast monolayer into two regions with different properties, that is actively dividing and growth arrest. The position of this boundary moved deeper into the colony as the external glucose concentration increased (*Figure 3d*).

We extended this idea further and used the complete HXT7 expression profile to infer the glucose concentrations at all positions within the chambers. Assuming that the local level of HXT7 expression is only set by the local glucose concentration, we can use batch culture measurements of HXT7 expression (based on flow cytometry) to determine the glucose concentration at a given chamber position (*Figure 4c and d*). However, this only allows us to reconstruct the glucose concentration gradient up to 0.016% *w/vol.*, that is in the domain where cells are actively dividing. The idea is simply to linearly map the two sets of measurements (in batch culture and in the microfluidic device) based on the fluorescence levels that correspond to the maxima $F_{max}$ and $F'_{max}$ and HXT7-GFP fluorescence levels at the chamber entry $F_0$ and $F'_0$. Using the data for HXT7 in *Figure 3*, we were able to reconstruct the glucose gradient for different initial glucose concentrations (*Figure 4e*). When applied to HXT1, the same inference led to very similar results (*Figure 4f*). In both cases, glucose concentrations decay very quickly moving away from the chamber opening and then exhibit a relatively long tail moving deeper into the colony.

## Gene expression landscapes of other genes and transcription factor activity confirm the inferred glucose gradients

The fact the seven glucose transporters exhibited varied, robust spatial expression patterns under identical conditions (*e.g.*, *Figure 4a*), together with the observed growth rate landscapes (*Figure 2*), suggests cellular metabolic state varies significantly across the longitudinal axis of the yeast monolayers. This variation was further assessed by mapping the expression and localisation of additional key genes involved in glucose metabolism.

MIG1 is a key transcription factor involved in glucose repression that localizes to the nucleus in the presence of glucose, to repress genes that participate in parallel carbon metabolic pathways (*e. g.*, galactose) (*Conrad et al., 2014*; *Broach, 2012*). Observing the cells at high magnification, we quantified the distance after which MIG1 fluorescence was not present in the nucleus of the cells (*Figure 5b*). This distance, around 400 µm at $C_0$ = 2% *w/vol* glucose, was in excellent agreement with the data obtained by HXT7 profiling. Interestingly, the spatial transition from nuclear MIG1 to cytoplasmic MIG1 localisation was very sharp and occurred over just a few cells.

In agreement with the batch culture observations, we found HXT5 was only expressed in regions with very low or no glucose concentrations where the cells did not seem to divide over several hours (*Figure 5a*). Therefore, HXT5 appears to be an excellent marker of growth arrest in this context (*Verwaal et al., 2002*).

The expression landscapes of two hexokinases involved in glucose metabolism, HXK1 and HXK2 (*Figure 5c*) that are expressed when cells are grown on non-glucose carbon sources, were also consistent with the batch measurements (*Figure 5d*, *Figure 5—figure supplement 3*) and further validated the existence of a glucose gradient. For each profile, we extracted the position of maximal expression and inferred the glucose concentration at this position from the FACS measurements of batch cultures. The batch measurements indicated maximal HXK1 and HXK2 expression were observed at a glucose concentration of about 0.016% *w/vol*. As expected, neither enzyme was expressed at very high glucose concentrations. The HXK1 and HXK2 expression maxima were similar at the two other glucose concentrations studied, around 300 µm at $C_0$ = 1% and 500 µm at 2% *w/vol*. Again, these data are in very good agreement with the positions of HXT7 peak expression at the same glucose concentrations.

Finally, we examined the expression of PDC1 and SDH2, which are overexpressed in fermenting and respiring cells, respectively (*Otterstedt et al., 2004*; *Bonander et al., 2008*; *Ohlmeier et al., 2004*). Their expression landscapes were inversely correlated (*Figure 6a*, *Figure 6—figure supplement 1*), indicating a transition from fermentative metabolic activity at the nutrient front of the colony to respiratory metabolic activity towards the dead end of the chamber where glucose is scarce. These expression maps are in good accordance with our previous results (*Figures 2*, *3* and *5*) and the levels of PDC1 and SDH2 expression in batch culture (*Figure 6b and c*).

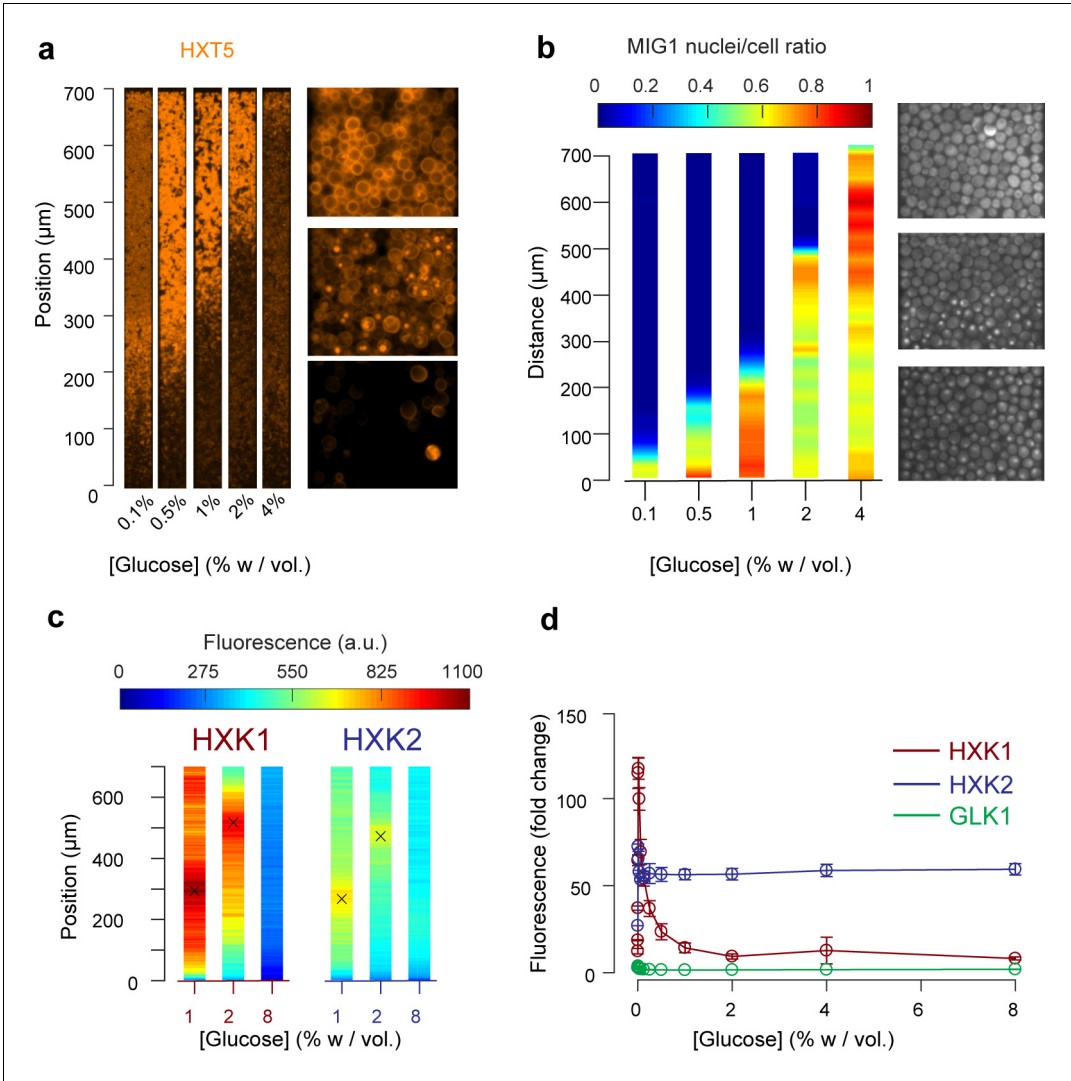

**Figure 5.** Other landscapes of genes involved in glucose metabolism. (**a**) Landscape of HXT5 expression. HXT5 is expressed under very low and no glucose conditions and appears to be a good marker of growth arrest. At $C_0$ = 2% *w/vol*, HXT5 expression is in good agreement with the observed absence of cellular division (see *Figure 2*, *Figure 5—figure supplement 1*). (**b**) Landscape of MIG1 activity. MIG1 fluorescence was located in the nucleus in the presence of glucose, with a sharp transition in nuclear localization observed (middle picture, at 2% *w/vol* glucose in the nutrient channel), confirming the existence of a glucose gradient (n = 3 replicates). Total number of cells and cells with nuclear localization of fluorescence were annotated manually and binned into 25 μm bins (see also *Figure 5—figure supplement 2*). (**c**) HXK1 and HXK2 are hexokinases involved in glucose metabolism. Their landscape of expression exhibited peaks that indicate a transition from high to very low glucose levels (n = 8–9 replicates per glucose concentration). (**d**) FACS measurements of HXK1 and HXK2 expression over a range of glucose concentrations (n = 3–6 replicates per glucose concentration).
DOI: https://doi.org/10.7554/eLife.47951.014

The following figure supplements are available for figure 5:

**Figure supplement 1.** Extended figure of *Figure 5a*.
DOI: https://doi.org/10.7554/eLife.47951.015

**Figure supplement 2.** Extended figure of *Figure 5b*.
DOI: https://doi.org/10.7554/eLife.47951.016

**Figure supplement 3.** Fluorescence intensity of key glucose metabolism genes measured by FACS over a range of glucose concentrations in batch culture.
DOI: https://doi.org/10.7554/eLife.47951.017

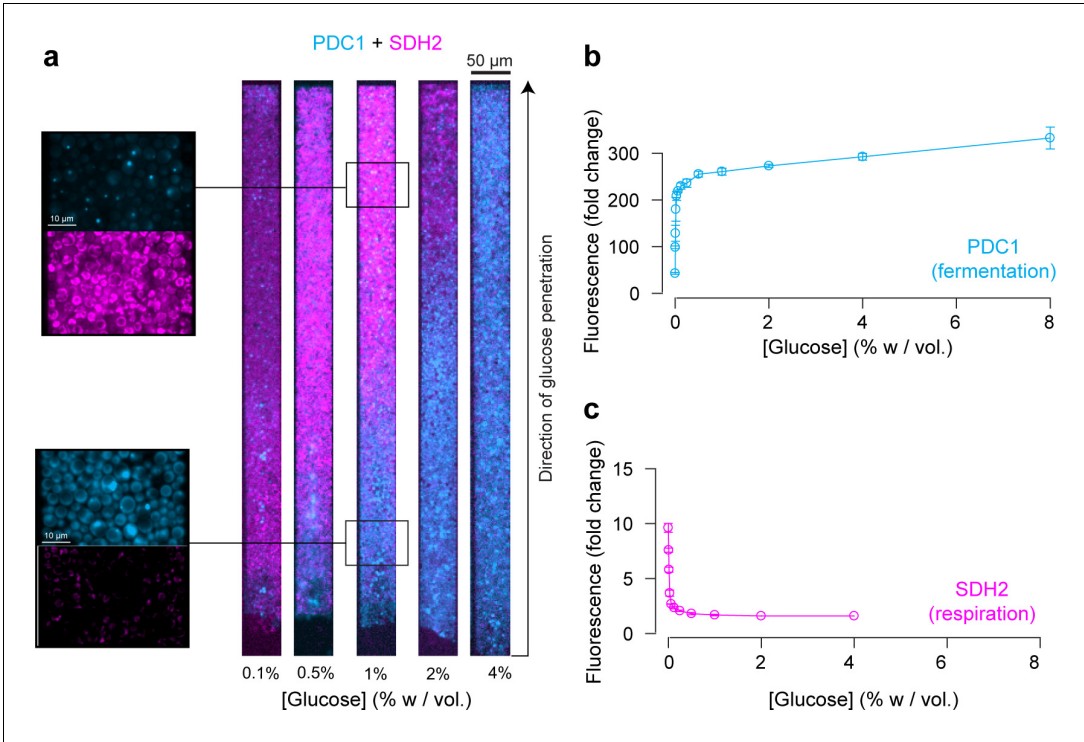

**Figure 6.** Impact of the glucose gradient on yeast physiology and the emergence of a landscape of phenotypes. (a) Overlay of the landscapes of gene expression of PDC1 (blue) and SHD2 (pink). PDC1 is known to be expressed when yeast cells ferment, SDH2 is mainly expressed in respiring cells (see also *Figure 6—figure supplement 1*). (b) FACS measurements of PDC1 expression over a range of glucose concentrations in batch culture (n = 3). (c) FACS measurements of SDH2 expression over a range of glucose concentrations in batch culture. The inverse correlation between PDC1 and SDH2 expression observed in batch culture is in good agreement with the inversely correlated spatial expression patterns within yeast cell monolayers (n = 3).

DOI: https://doi.org/10.7554/eLife.47951.018

The following figure supplement is available for figure 6:

**Figure supplement 1.** Extended figure of *Figure 6a*.

DOI: https://doi.org/10.7554/eLife.47951.019

## Multiple gene expression landscapes are spatially correlated

We decided to compare the landscapes of gene expression for the entire set of reporter genes by aligning the different landscapes across varied nutrient conditions (*Figure 7a*). Strikingly, all landscapes showed a high level of spatial correlation. Two major landscapes emerged: peaking (*e.g.*, HXT7) and switching (*e.g.*, HXT1 or MIG1). We defined and extracted the typical lengths of the peaking and switching landscapes (*Figure 7b*) and plotted them as function of the external glucose concentration (*Figure 7c*). The typical lengths of all of these landscapes for different reporter genes were remarkably close, despite the fact that we looked at different cellular components: a transcription factor (MIG1), glucose transporters (HXTs), metabolic enzymes (HXKs) and metabolic state reporters (SDH2, PDC1). Notably, we gained a global view of gene expression landscapes and their interrelationships along a monolayer colony. All data showed the colonies were structured into two regions with very different properties (*Figure 7d*): an actively growing region, where cells divide abundantly and ferment glucose, and a quiescent area, where cells do not divide much and have switched to respiratory metabolism to compensate for the very low glucose availability. While it is not surprising to see the expression levels of metabolic genes vary with the glucose concentration, our approach demonstrates genetic programs not only allow individual cells to adapt to changes in the nutrient environment, but also enable multicellular assemblies to spatially self-organize through long-range metabolic interactions governed by physical rules of diffusion and uptake. This sheds new light on the coordinated actions of these genes in individual cells in a biologically relevant

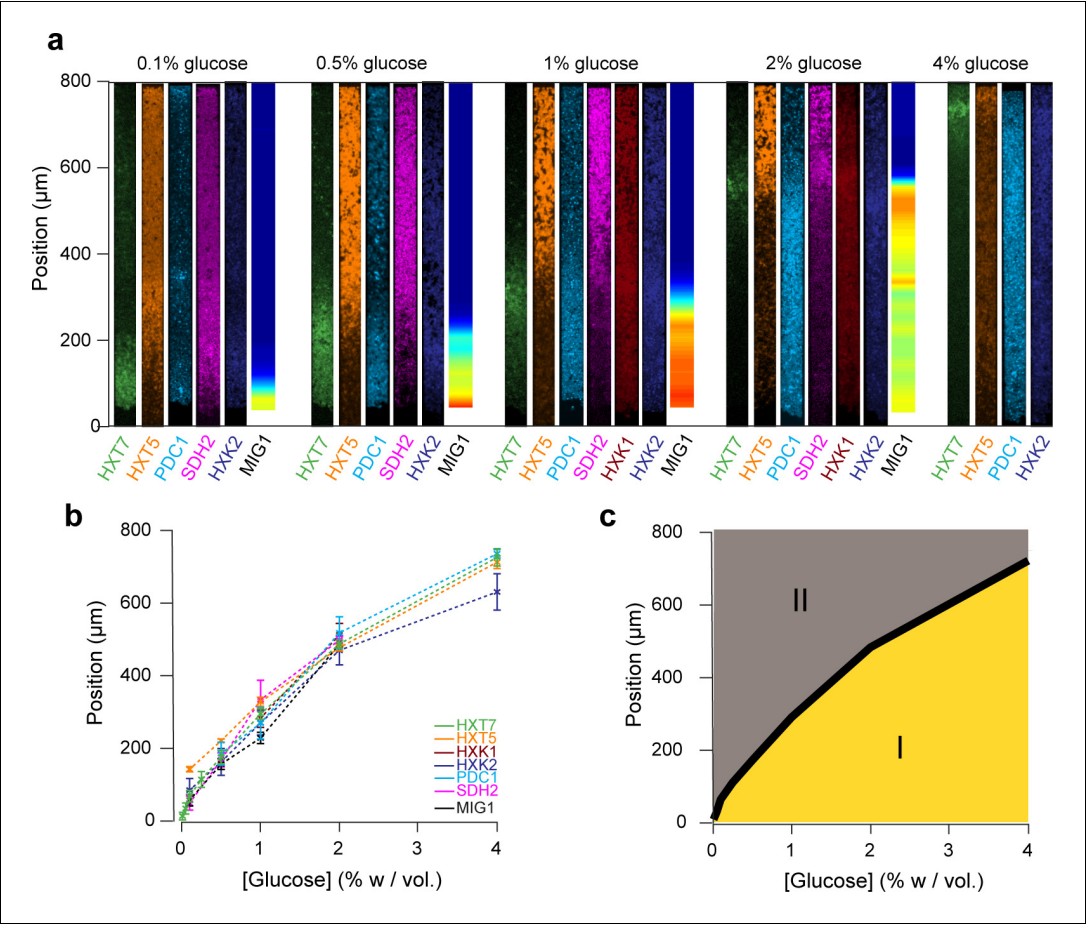

**Figure 7.** Global view of the emergence of landscapes of gene expression. (**a**) The different landscapes of gene expression presented in this study are aligned, regrouped and displayed over a range of glucose concentrations. This simple view sheds light on the macroscopic spatial correlations between these different landscapes, which are both setting and traces of the establishment of glucose gradients. (**b**) For each gene expression landscape, we identified the fluorescence peak (HXT7, HXK1, HXK2) or the position of the transition between low and high expression (HXT1, HXT5, SDH2, PDC1) or activity of the transcription factor (MIG1). (**c**) Landscapes of gene expression delimit two regions in which cells are physiologically different. Phase I indicates active growth by fermentation in the presence of glucose; Phase II indicates growth arrest or very limited growth via respiratory metabolism at zero or close to zero glucose concentrations. The transition between the two phases typically takes place relatively sharply, over a hundred micrometers or ~20 cells.

DOI: https://doi.org/10.7554/eLife.47951.020

multicellular context that has impact on ecology, evolution, development and emergence of multicellularity.

Overall, we studied how cells within a monolayer colony collectively shape their microenvironment through long-range metabolic interactions. This is a complex process, in which individual cells adapt locally, and shape a spatial landscape of gene expression as a global phenotype. As a whole, the structure of an assembly of cells and the microenvironment landscapes emerge as the result of local cellular metabolic activity of individual cells.

## Discussion

Here, we took an alternative point of view compared to traditional systems and single-cell biology. Rather than studying single-cell metabolic properties in a well-mixed, homogeneous environment, we designed a microfluidic chip to force yeast cells to grow and shape their microenvironment, solely by fixing the properties of the microenvironment at the boundary of the monolayer. This

approach allowed us to measure simultaneously properties at both the single-cell scale and structured population scale and holds potential for establishing a quantitative link between these scales.

In standard batch culture, as cells exhaust the media, their adaptation time, limited by sensing, transcription and translation, may lag behind the decrease of glucose in the media. In contrast, in our 2D colony device, once steady-state gradient is settled, cells' residence time within a given range of glucose that corresponds to stable HXT expression is significantly longer, assuring that even though cells are continuously pushed away, they spend sufficient time within a given concentration to equilibrate their response to the gradient and therefore faithfully report the glucose concentration in their environment. As example, at 1% glucose the mean velocity at Hxt7 expression peak, spanning over >200 μm (*Figure 3a*, varies between 5–15 μm/hour (*Figure 2f*), suggesting cells' residence time of >10 hr, pointing to the advantage of working in this setup.

Specifically, we showed that cells self-generate nutrient landscapes that in turn influence cellular metabolism and gene expression profiles. This behavior, based on nutrient uptake adaptation, is generic and feeds back on the behavior of other cells through what we call non-specific long-range metabolic interactions. Indeed, the microenvironment sensed by cells a few hundred micrometres inside a colony is very different from the microenvironment experienced by the cells at periphery. The resulting patterns may emerge from the individual adaptive properties of the cells without the need to evoke specific higher-level community properties as cell-cell communication. Notably, gradients emerge over relatively short distances, and this process may possibly affect studies of cellular populations within microfluidics settings. More importantly, quantitative description of gene expression landscapes is critical if one wants to understand the establishment and behavior of cellular populations, whether these are as simple as yeast colonies or more complex, such as biofilms and complex microbial ecosystems in which several types of cells cohabit and may specifically communicate and interact. Indeed, in addition to the described long-range metabolic interactions, many other environmental and genetic determinants such as intercellular communication, cell surface properties, cell-cell adhesion strength and secretion of extracellular matrix components have been shown to participate in the emergence of the complex morphology (*Nadell et al., 2016*; *Granek and Magwene, 2010*; *Flemming et al., 2016*) and internal structure of microbial colonies in such complex situations. The nature of many of these interactions could also be studied using similar microfluidic devices to identify the relative contribution and relationship of environmental and genetic determinants to the metabolically generated microenvironment.

We have made notable advances in the study of emerging properties of yeast colony growth, microenvironment formation and gene expression compared to previously published studies (*Cáp et al., 2012*; *Maršíková et al., 2017*; *Palková et al., 2014*). These studies have shown fascinating differentiation and diversity within yeast colonies grown on agar but their relevance to study the dynamical emergence of complexity in microbial colonies is limited by their methodology (*e.g.*, growth on a single specific medium with no dynamic control of environmental changes, two-photon microscopy, unsuitable for live time-lapse microscopy, obligation to section colonies *etc*.) which does not allow detailed spatiotemporal analysis of cellular growth, microenvironment and gene expression landscapes at a relevant single-cell scale. Our approach is designed to access the dynamics of large microbial colonies, and while we did not report it here, it is straightforward with microfluidics to dynamically change in frequency and composition the external environment, and as such to analyse how colonies adapt their internal organization to such stresses.

Our results are in most part in line with the knowledge of glucose metabolism obtained in batch culture. Yet, our methodology sheds quantitative description of the spatial expression of genes involved in the glucose metabolism and its correlation with the cell local growth rates. Our results show that even in the simple context studied here, reconstructing the microenvironment spatial structure from single-cell measurement is not trivial. A proper model should take into account how the growth rate and specific absorption rate vary with the glucose concentration and the microenvironment. Modeling the entire complexity of the microenvironment is hardly possible, even with today's knowledge. Thus, we decided to take a different approach and use key genes involved in glucose metabolism to infer the glucose concentration gradient. We showed that different reporter genes consistently reported the same glucose gradient. We envision that the data extracted from relevant fluorescent reporters could be fed into an agent-based or mean-field models that take cell-cell interactions, mechanics and spatial diffusion of metabolites into account to fill the gap between data generated from single cells to data that is relevant to evolution and ecology, that is at the

colony scale. We anticipate that linking local properties to macroscopic, global behavior will help to understand the architecture of microbial communities and how evolution shapes the development of these architectures through long-range metabolic interactions and possible inherent biological programs that coordinate it. Of note, in another rare attempt to study emergence of population level phenomena in yeast *S. cerevisiae* *Campbell et al. (2015)* looked at the synthetic 'self-establishing communities' that were able to cooperatively exchange metabolites. They inoculated on agar plate auxotrophic *S. cerevisiae* strain that had different auxotrophic markers on plasmids. As cells were dividing, some of the plasmids that complemented yeast auxotrophy and therefore rescued their growth were lost, resulting in a colony which is composed of yeast that are auxotrophic for a certain amino acid. However, they were able to grow because they used amino acids that were released in the environment by other yeast that were producing it, effectively generating a very heterogeneous colony that sustained growth through metabolite exchange. Interestingly, previous efforts to co-culture complementary auxotrophs had limited effectiveness in supporting co-growth in liquid cultures, indicating the importance of spatial structure in facilitating cooperation and makes our system very attractive for study of such phenomena (*Wintermute and Silver, 2010*).

Furthermore, while the spatial microenvironment is not fully characterized, we have shown that the emergence of gradients, and simultaneously gene expression landscapes, are robust and reproducible features of the colony. Moreover, the landscapes can be compared to extract correlation patterns and infer how gene regulatory networks act in synchronicity to establish the microenvironment within the colony. This approach may provide a relatively simple, yet effective method of screening for 'organismic' properties that have been shaped by evolution and are only relevant in a multicellular context.

Our future efforts to extend the application of this setup will be dedicated to the study of how the microenvironment dynamically changes when external conditions are altered, an uncharted territory at the scale of a multicellular assembly that is central to the understanding of microbial ecosystem resistance to stress, environmental fluctuations and adaptation. We anticipate that similar approaches could be used to study aging, cooperation and competition, cell memory or evolutionary dynamics, as well as quantitative characterization of (synthetic) ecological systems and mixtures of cells relevant to ecology and chemical biology.

## Materials and methods

### Yeast strains

All experiments were performed using haploid *S. cerevisiae* strains derived from the S288C background - BY4741: *MATa his3Δ1 leu2Δ0 met15Δ0 ura3Δ0*. See *Supplementary file 1* for a detailed list of the yeast strains used in this study.

### Microscopy

We used an inverted fluorescence microscope (IX81, Olympus) equipped with an EMCCD camera (Evolve 512, Photometrics) and X-Cite exacte fluorescence light source (Lumen Dynamics). Optical filters from Chroma Technology Corporation ET-EGFP (U-N49002; Ex 470/40 nm Di495 Em 525/50 nm) and ET-DsRed (U-N49005; Ex 545/30 nm Di570 Em620/60 nm) were used to observe GFP and RFP fluorescence. Cells were observed using Olympus 10× (Plan 10x/0.25 NA), 60× (PlanApo N 60x/1.42 NA Oil) and 100× (UPlanFL N 100x/1.3 NA Oil) objectives. Open-source μManager (*Edelstein et al., 2014*) microscopy software was used to control all of these components and setup multi-dimensional acquisition. The temperature inside the microscope incubation chamber that contained the media and cells was maintained at 30°C (Life Imaging Services). Fluorescence intensity was set to 10% of maximum output, fluorescence exposure was set to 1000 ms and camera gain was set at maximum. The time interval between each acquisition cycle was 6 min.

### Microfluidics and cell loading

Microfluidic devices were constructed using soft lithography techniques. Photomasks were drawn using L-Edit software (Tanner) and printed on a high-resolution glass substrate (Delta Mask). A master wafer was created using SU-8 2000 (MicroChem) epoxy-based photoresist that was spin-coated to the appropriate thickness and exposed to UV light using an appropriate photomask to create the

desired pattern. Multi-layered patterns were aligned and exposed to UV light using a MJB4 manual mask aligner (SUSS MicroTec) and the dimensions of the master wafer were checked using a Dektak 150 surface profiler (Veeco). The master wafer was treated with 95% (3-mercaptopropyl)-trimethoxy-silane (Sigma) for 1 hr in the vapor phase. Microfluidic chips were created by casting a degassed 10:1 mix of polydimethylsiloxane (PDMS) and curing agent (Sylgard 184 kit; Dow Corning) on the master wafer, followed by at least 2 hr curing at 65°C. Each chip was gently cut and peeled off the master wafer; the entry/exit ports were punched out. The chip and a glass coverslip (24 × 50 mm #1; Menzel-Gläser) were treated with $O_2$ plasma for 1 min in a plasma cleaner (Harrick Plasma), bonded together and incubated at 65°C for 10 min. Before loading cells, the chips were coated with 1% Pluronic F-127 (Sigma) for 30 min. Cells were precultured overnight in 5 mL of synthetic complete (SC) medium containing 2% *w/vol* glucose in a shaking incubator at 30°C, diluted 50-fold into 50 mL of SC +2% *w/vol* glucose, cultured for 5–6 hr in a shaking incubator at 30°C to an $OD_{600}$ of 0.2–0.4, collected by centrifugation, and loaded into the microfluidic system with a pipette. The microfluidic system was centrifuged for 2 min at 1000 rpm using 3D-printed adaptors (Laurell WS-650 spin coater) to force the cells into the dead-end cell chambers. Liquid media was flowed rapidly through the flow channel to remove excess cells and the flow rate was set to 5 µL/min. A pressure-based microfluidic flow control system (MFCS; Fluigent) coupled with a flow rate platform (Fluigent) and a flow rate control module (Fluigent) that measured the flow rate and kept it constant by adjusting the pressure through a feedback loop was used to push liquid media through the flow channel. The output was kept at a constant pressure of 100 mbar above atmospheric pressure to minimize formation of air bubbles inside the flow channel.

## Flow cytometry

Flow cytometry experiments were performed on a Gallios Flow Cytometer (Beckman Coulter) using a 488 nm excitation laser and 530/30 nm FL1 emission filter to detect GFP fluorescence. Data analysis was performed using Kaluza Flow Cytometry Analysis Software (Beckman Coulter). Approximately $10^4$ cells were inoculated in 10 mL of SC medium containing various glucose concentrations ($\log_2$ dilutions from 8% to 0.0078125%, and 0% *w/vol* glucose) and cultured in a shaking incubator at 30°C to an $OD_{600}$ of ~0.02–0.2 depending on the starting glucose concentration. Cells were then diluted 10-fold into 10 mL of fresh SC media containing the same starting glucose concentration and grown for 4–5 hr in a shaking incubator at 30°C, centrifuged at 4000 rpm for 10 min, re-suspended in 300 µL of PBS pH 7.4 buffer (Gibco) and fluorescence was measured using the flow cytometer. The supernatant of each sample was collected, and the glucose concentration was measured using the Glucose (HK) Assay Kit (Sigma) to confirm that the glucose concentration remained at similar levels during the growth phase (*Figure 5—figure supplement 3*). This is a modification of a previously published protocol (*Youk and van Oudenaarden, 2009*) for measuring expression of glucose transporters in different glucose concentrations modified to minimize glucose depletion in batch culture prior to expression measurements by working with sufficiently diluted batch populations. We thus expect that these measures represent expression levels that correspond to the starting glucose concentration, and would therefore be close to the measure within the 2D colonies (see discussion).

## Image analysis

Image analysis was performed using open-source ImageJ 1.51 p software (*Schneider et al., 2012*). To obtain front velocity, we applied a threshold (Otsu) to detect the bottom frontier over time after flattening the background using a FFT band-pass filter. The image signal is decomposed by FFT into a spectrum of its constituent frequencies. Because some operations can be more easily performed on the spectrum than on the original image, the FFT bandpass algorithm filters out large structures (shading correction) and small structures (smoothing) of the specified size by gaussian filtering in Fourier space. The default parameters are set at 40 pixels for large structure and five pixels for small ones. To compute the local speed of the cells inside the cell assembly, we used the plugin Track-Mate (*Tinevez et al., 2017*) v3.5.1 to track cell trajectories. TrackMate was set to DoG detector with estimated blob diameter of 4 µm and threshold of 4, while tacking was set to linear motion LAP.

## Acknowledgements

The authors would like to thank Sébastien Léon (IJM, CNRS) and their respective team members for their critical reading of this manuscript. This work was supported by the *Agence Nationale de la Recherche* (ICEBERG-ANR-10-BINF-06–01; ANR-16-CE12-0025-01), the interdisciplinary program of the *University Sorbonne Paris Cité*, the *Who am I?* Laboratory of Excellence (ANR-11-LABX-0071 and ANR-11-IDEX-0005–01) and the European Research Council (ERC) under the European Union's Horizon 2020 research and innovation programme (grant agreement No 724813).

## Additional information

### Funding

| Funder | Grant reference number | Author |
| --- | --- | --- |
| European Commission | 724813 | Pascal Hersen |
| Agence Nationale de la Recherche | ICEBERG-ANR-10-BINF-06-01 | Pascal Hersen |
| Agence Nationale de la Recherche | ANR-16-CE12-0025-01 | Pascal Hersen |
| Agence Nationale de la Recherche | ANR-11-LABX-0071 | Pascal Hersen |
| Agence Nationale de la Recherche | ANR-11-IDEX-0005–01 | Pascal Hersen |

The funders had no role in study design, data collection and interpretation, or the decision to submit the work for publication.

### Author contributions

Zoran S Marinkovic, Data curation, Investigation, Writing—original draft, Writing—review and editing; Clément Vulin, Investigation, Writing—original draft; Mislav Acman, Visualization, Methodology; Xiaohu Song, Software, Image analysis; Jean-Marc Di Meglio, Ariel B Lindner, Supervision, Writing—original draft, Writing—review and editing; Pascal Hersen, Conceptualization, Supervision, Funding acquisition, Writing—original draft, Writing—review and editing

### Author ORCIDs

Zoran S Marinkovic https://orcid.org/0000-0002-7720-2802
Clément Vulin https://orcid.org/0000-0003-3914-0708
Jean-Marc Di Meglio http://orcid.org/0000-0003-0446-4550
Ariel B Lindner https://orcid.org/0000-0001-5015-6511
Pascal Hersen https://orcid.org/0000-0003-2379-4280

### Decision letter and Author response

Decision letter https://doi.org/10.7554/eLife.47951.026
Author response https://doi.org/10.7554/eLife.47951.027

## Additional files

### Supplementary files

• Supplementary file 1. List of yeast *S. cerevisiae* strains used in this study.
DOI: https://doi.org/10.7554/eLife.47951.021

• Transparent reporting form
DOI: https://doi.org/10.7554/eLife.47951.022

### Data availability

All data presented in the figures, article, and supplementary information are available on Zenodo (http://dx.doi.org/10.5281/zenodo.2557396).

The following dataset was generated:

| Author(s) | Year | Dataset title | Dataset URL | Database and Identifier |
|---|---|---|---|---|
| Marinkovic Z, Vulin C, Acman M, Song X, Di Meglio JM, Lindner AB, Hersen P | 2019 | Lab513/Landscape of Gene Expression Dataset | http://dx.doi.org/10.5281/zenodo.2557396 | Zenodo, 10.5281/zenodo.2557396 |

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
