## [Decision Letter]

[Editors’ note: a previous version of this study was rejected after peer review, but the authors submitted for reconsideration. The first decision letter after peer review is shown below.]

Thank you for submitting your work entitled "Emergence of metabolic landscapes in yeast monolayer colonies" for consideration by *eLife*. Your article has been reviewed by three peer reviewers, and the evaluation has been overseen by a Reviewing Editor and a Senior Editor. The reviewers have opted to remain anonymous.

Our decision has been reached after consultation between the reviewers. Based on these discussions and the individual reviews below, we regret to inform you that your work will not be considered further for publication in *eLife*.

We appreciate how your study uses a microfluidics setup to study the behavior of *S. cerevisiae* in a glucose gradient. The combination of this setup with GFP tagged version of hexose transporters, a glucose sensitive transcription factor (MIG1), and other reporters of metabolic state (PDC1, SHD2) allows studying spatial patterning as a function of glucose availability. The results show that patterns of expression of hexose transporters with different affinities are spatially arranged in a manner consist with a gradient of glucose availability with highest amounts of glucose available near the "source" (i.e. periphery of the artificial colony), and lowest levels near the "sink" (i.e. center of the artificial colony). Perhaps more interestingly, interpretation of the data suggests a relatively sharp transition in the colony between zones of growth and slow/no-growth, which they ascribe to zones of fermentative and aerobic metabolism.

Overall, the reviewers agree that this is a particularly elegant study. As you can see in the individual reports (below), reviewer 1 is significantly more positive than reviewers 2 and 3. However, after discussing the individual reviews, all reviewers question the amount of novel biological insight that is presented in your paper.

As such, I am afraid that we are not able to offer publication in *eLife* (unless of course you believe that we missed the major message and its biological relevance). On the other hand, everyone also agrees that the experiments and results are beautiful, and that the setup may be usable for other studies. As such, we wonder whether it would be conceivable to re-write your paper as a "Tools and resources" paper? If so, you would need to emphasize the design and operation of the system, detail its use for a broad range of studies and compare it to other, existing systems such as the ones describes by Hornung et al., 2018, and Wilmoth et al., 2018. The Hornung et al. paper in particular seems to use a rather similar setup…

*Reviewer #1:*

The authors applied a microfluidic device to study how long-range metabolic interactions influence the internal structure of simple yeast cell monolayers asymmetrically exposed to an external nutrient source. They showed spatial variation in nutrient levels, cellular growth, and expression of several metabolic genes to self-emerge. Notably, the gene expression landscapes exhibited a high degree of spatial correlation over different levels of external nutrient level and were indicative of a clear spatial transition between fermentative and respirative growth.

I really enjoyed reading this work. Although the experimental set-up is simple and the conclusions intuitive and straightforward, the depth of the analyses and the clarity of the interpretations are impressive. The authors combined growth measurements, single cell tracking, fluorescence based expression analysis with FACS and microscopy and advanced data representation to study spatial relations between nutrients, growth and metabolism with an unprecedented level of detail. This work will therefore undoubtedly provide inspiration to many other researchers interested in spatial heterogeneity of microbial communities. I therefore recommend publication in *eLife*.

*Reviewer #2:*

Marinkovic et al. address single-cell response to glucose gradients within structured communities. They present an intriguing method to look at metabolic landscapes in a microbial community. However, I do believe the novelty of the results need to be clarified significantly better, and the conclusions drawn must be discussed more critically. Alternatively, the work may be more suitable for a methods focused publication, as using the gene expression of the transporters to gauge the extracellular concentration of glucose within a colony is an interesting tool.

Major concerns:

A) The glucose gradients emerge in the end artificially, due to the design of the microfluidic chamber and due to the fact the yeast's consume glucose, and not due to community action. The method of the author's pictures of how the cells respond to different glucose availability as a function of glucose concentration in different areas of the closed channels. But the effects are the same if one cultivates the cells in batch cultures, and tunes glucose availability in this way. One does hence not learn anything about yeast communities, only, that yeast in batch culture and in a microfluidic chamber respond similarly to glucose depletion. Hence, is there anything new in the study, other than nicely illustrating the well-established fact that yeast activates different glucose transporter at different glucose concentration?

B) Batch cultures are used to tune gene expression sensors. But, in batch culture, glucose is constantly and rapidly depleted, while gene expression, needs some time to adjust. At any given time, in batch culture, the glucose concentration in the media, and the gene expression programme, are hence asynchronous (i.e. gene expression lags behind the glucose concentration). Chemostat experiments would be needed, to have a glucose concentration that corresponds to the gene expression programme at a particular point in time.

Method:

Figure 1: I have not found the mention of the exact number of dead-end chambers within one microfluidic chip (I'm guessing 16 from the figures?). However, looking at the n in different experiments (e.g. Figure 1, n=12, n>5 and n=9) I am wondering in how many separate experiments these were done, and how much of the collected data was discarded and for which reasons?

Was it excluded that nutrient concentrations from the previous chamber affected the later ones? Especially in the stages when the chambers were already filling up/cells were being carried away?

Results and Conclusion:

Subsection “Cellular metabolic activity creates gene expression landscapes”: "Both patterns demonstrate the formation and maintenance of a glucose gradient that emerges from cellular metabolic activity." The pattern of the gene expression demonstrates that the cells adapt to the glucose gradient that emerges through supply of fresh media from only one side. "Formation and maintenance by cellular metabolic activity" in the sense that, yes, the cells close to the opening do import the glucose they require seems a weak point to make. In other words, there is evidence for the contrary – there seems no different action of cells in the community as in the batch culture, when it comes to the expression of the transporters.

Discussion paragraph four: What is the dependence/independence of the different gene expression looked at here? Can we talk about synchronicity between networks if the studied genes are known to dependent on the same external factor?

Figure 6: Landscapes of gene expression is not shown at all glucose concentrations for all genes, why?

*Reviewer #3:*

This manuscript details the results of a study using a microfluidic system to study growth and gene regulation in monolayers of yeast cells. This system is presented as a tractable model for studying pattern formation in microbial colonies, via spatial and temporal measurement of cell division rates and measurement of gene expression via fluorescently tagged reporters.

The authors use this experimental setup, in combination with GFP tagged version of hexose transporters, a glucose sensitive transcription factor (MIG1), and other reporters of metabolic state (PDC1, SHD2) to study spatial patterning as a function of glucose availability.

Unsurprisingly, they find that patterns of expression of hexose transporters with different affinities are spatially arranged in a manner consist with a gradient of glucose availability with highest amounts of glucose available near the "source" (i.e. periphery of the artificial colony), and lowest levels near the "sink" (i.e. center of the artificial colony). Perhaps more interestingly, their interpretation of their data suggest a relatively sharp transition in the colony between zones of growth and slow/no-growth, which they ascribe to zones of fermentive and aerobic metabolism.

While I appreciate the elegance of the microfluidic colony model that is presented, the key findings are rather modest -- i.e. there are gradients of nutrient availability in microbial colonies. This doesn't really have the impact I expect from *eLife* papers.

Below I detail a number of concerns/comments about the study, the analyses, and whether the authors have adequate related their work to the larger literature in this area.

Detailed comments:

1) Several other recent papers (Hornung et al., 2018,; Wilmoth et al., 2018) have used microfluidic setups to study microbial colonies. The Hornung et al. paper is the most similar to the current experimental design.

2) If goal was to measure the rate of glucose uptake and availability it seems like there may be other more direct approaches than inferring this from hexose transporter expression, such as the use of 2-NBDG, a non-metabolizable, fluorescent glucose analog (Roy et al. 2015, doi: 10.1371/journal.pone.0121985).

3) The following paper, while not at the fine spatial scale provided by the microfluidics, presents a more complete and compelling view of metabolic differentiation in yeast colonies: Maršíková et al., 2017.

4) The spatial analyses focus primarily on peaks of maximum expression, but the data is potentially much richer and more interesting if the authors consider not only the global maxima but more complex spatial patterns. For example, in Figure 4B MIG1 nuclear localization appears to be multimodal. What is going on here?

5) There is very little effort to contextualize this work in the larger body of studies of yeast colony structure, physiology, or regulation of glucose responsive TFs. Illustrative of this, there are only three citations total in the entire discussion. There is only a single reference to the regulation of hexose transporters in the entire manuscript. Specific examples:

- Subsection “Cellular metabolic activity creates gene expression landscapes”: " HXT1 is a low-affinity glucose transporter mainly expressed under high-glucose conditions, while HXT7 is a high-affinity glucose transporter expressed under low-glucose conditions only" Citations?

- Subsection “Gene expression landscapes depend on the glucose source concentration”: "As HXT1 is mainly expressed under high-glucose conditions (> 1% w/vol glucose) in batch culture…" Citation?

- Subsection “Gene expression landscapes of other genes and transcription factor activity confirm the inferred glucose gradients”: "MIG1 is a key transcription factor involved in glucose repression that localizes to the nucleus in the presence of glucose, to repress genes that participate in parallel carbon metabolic pathways (e.g., galactose)." Citation?

- and in the same subsection: "…we examined the expression of PDC1 and SDH2, which are overexpressed in fermenting and respiring cells, respectively" Citation?

[Editors’ note: what now follows is the decision letter after the authors submitted for further consideration.]

Thank you for resubmitting your work entitled "A microfluidic device for inferring metabolic landscapes in yeast monolayer colonies" for further consideration at *eLife*. Your revised article has been favorably evaluated by Naama Barkai (Senior Editor), a Reviewing Editor, and three reviewers.

We really appreciate the efforts of re-writing this manuscript as a "resource" paper. We feel that the paper has been much improved, but there are some remaining issues that need to be addressed through textual changes before acceptance, as outlined below:

1) We ask that the authors make it clear to the reader that what they call a "community action" does not imply that cells are responding differently in the community than they would as single cells in planktonic growth. The observed effects (gradients and response to these gradients) are merely the consequence of a high concentration of more or less static cells. So, perhaps calling it "population effect" would be more appropriate?

2) We would appreciate if you could insert a succinct discussion of the difference between a batch culture and this steady-state (where in principle, the cells could be better adapted as they are not always "running behind" in their response to a variable environment, as they would be in batch). (see reviewer 2's comments below for more details)

3) We again ask to include more references and to not overstate the advantage of the presented device (see reviewer 3's comment below).

4) Please provide more details about the reproducibility/variability and statistics of the results and analyses.

The above points summarize what we believe to be essential changes, and we only ask for textual changes instead of further experiments. We are including the full reviews for your reference below.

*Reviewer #1:*

In this revised manuscript, the authors – as requested- presented their work as a tool that can be generally applied to quantitatively and dynamically study growth landscapes, metabolic landscapes and gene expression landscapes within extended monolayers of cells. They extensively elaborated on the similarities and differences of their approach compared to other studies that applied agar colony models or microfluidics devices to study spatiotemporal patterns in colonies, and clearly pinpointed advantages of their approach. A figure was added to clarify the experimental set-up. Also my other (minor) comments were addressed to a sufficient extent. I therefore recommend publication.

*Reviewer #2:*

Although the authors have put efforts in their revision, they have not addressed my two main points. I'm hence wondering if I have not expressed myself clearly enough, so it's perhaps my own fault – or it also could be a different use of language between disciplines, so that the authors did simply not understand my questions. Long story short, I don't want to sound negative as it may be my own use of language, but the revision has not addressed the two simple concerns I had. Perhaps the other reviewers could comment if they agree or disagree with me, I'm happy if my comments are ignored, in case my concerns are not shared.

I think both points still are relevant, though. The first one could be fixed simply in writing. 'Community action' implies to me, and I guess to many others in the field, that there are interactions between the cells that emerge in the community – implying that cells would behave differently in the community as if they are not in the community. When it comes to glucose consumption, this seems not to be the case – there seems nothing which indicates that the individual cell's glucose import would be depending on interaction terms between the cells. The manuscript is written as if glucose gradients would emerge because of 'community action', but in fact, they emerge from cells that consume the glucose, in a way that does not require a different action as if cells would act independently to one another. In other words, the cells deplete the glucose in the MF channels, and within the channel, this creates a gradient, to which the cells respond. This alone does not indicate that cells act differently in the community. as they would act in Isolation.

The second point may remain relevant, however. The sensors are tuned in batch culture. But in batch culture, one expects a time differential, between the glucose concentration present at a point in time, and the average gene expression program activated at that same moment (as gene expression needs time to adjust, but glucose is constantly depleted. I.e. if gene expression needs 30mins to adjust to a changed glucose concentration, the gene expression program reflects the glucose concentration that was in the batch 30mins earlier before the timepoint was taken). I had suggested the authors to control this in chemostats, where gene expression and concentration are a time-wise aligned. I agree this would have been time-consuming to do and could hence be difficult for the authors to do. But it would have been great if they could have come up at least with some idea, and quantify how big this effect is; One would assume the differential between glucose concentration and gene expression are strongest in the early and late exponential phase. If chemostats are too difficult, one could suggest also much simpler surrogate experiment, for instance. One could, for instance, re-supplement exhausted media with glucose, perform a time-course, and estimate in this way the time needed until the gene expression has re-adjusted to the supplied glucose level. The sensor data would need to be adjusted by the time differential.

*Reviewer #3:*

This revised manuscript has been appropriately rewritten as a "Tools and Resource" article.

1) The authors overstate the uniqueness of their approach. For example, in the Introduction in reference to previous studies using similar microfluidics designs:

" While the use of microfluidics gave rise to the discovery of interesting collective properties of microbial assemblies, such attempts were too specific and had to deal with some of the limitations like small device dimensions (<100 100 μm), use of low nutrient concentrations (<1 mM), limited scope of nutrient types, inability to access single cell level – and therefore cannot be transposed to the general case of the study of a large monolayer of cells in standard range and scope of nutrients employed in biological research."

Few of these critiques hold up. For example, low nutrient concentrations and limited scope of nutrient types are simply specifics factors applied to the different organisms study. This critique applies equally to the system presented here. Furthermore, the Hornung et al. article certainly allows access to the single cell level. The text should be revised to reflect a more accurate summary of this earlier related work.

2) The rebuttal states "We agree that more contextualization was necessary so we expanded our Introduction and Discussion part, as well as provided additional citations, including all the ones proposed by the reviewer" I can find no references our discussion of prior literature at the line numbers provided.

3) One of the points the authors make several times regards the reproducibility of their assays. For example: "We demonstrate a novel capacity to reproduce and

quantify…" and "…we found the growth pattern was highly reproducible across parallel chambers.." However in the *eLife* "transparent reporting form" the authors state "We did not use statistical inference or comparison between data sets." How then did the authors estimate reproducibility/repeatability?

---

## [Author Response]

[Editors’ note: the author responses to the first round of peer review follow.]

We thank the editors and reviewers for their careful review of our work. We concur that our work does not provide novel genetic network mechanistic insights but reveals and quantifies time- and length-scales at play in the emergence and the maintenance of gene expression and cellular growth patterns in a growing multicellular assembly. It also provides an elegant and straightforward method for quantitative spatiotemporal exploration of microbial colonies, as you outlined in your editorial decision. As such, and following your recommendations, we rewrote our article as a “Tools and Resources” submission, outlining the methodology and discussing how this could be used to gain novel insights in a broad range of multicellular systems. We changed the title, the abstract, rewrote part of the introduction and discussion sections and added one figure with methodological details of the microfluidic system and a few supplementary figures.

Reviewer #1:

The authors applied a microfluidic device to study how long-range metabolic interactions influence the internal structure of simple yeast cell monolayers asymmetrically exposed to an external nutrient source. They showed spatial variation in nutrient levels, cellular growth, and expression of several metabolic genes to self-emerge. Notably, the gene expression landscapes exhibited a high degree of spatial correlation over different levels of external nutrient level and were indicative of a clear spatial transition between fermentative and respirative growth.I really enjoyed reading this work. Although the experimental set-up is simple and the conclusions intuitive and straightforward, the depth of the analyses and the clarity of the interpretations are impressive. The authors combined growth measurements, single cell tracking, fluorescence based expression analysis with FACS and microscopy and advanced data representation to study spatial relations between nutrients, growth and metabolism with an unprecedented level of detail. This work will therefore undoubtedly provide inspiration to many other researchers interested in spatial heterogeneity of microbial communities. I therefore recommend publication in eLife.

We do thank the reviewer for his positive assessment of our work. In light of the other reviewers comments, we recall here that our goal was indeed to get a quantitative analysis of cellular growth and gene expression landscapes in an extended monolayer of cells. We did not expect to gain novel mechanistic insights on, say, the glucose genetic regulatory network, which has been studied in details in the past. On the contrary, we used this knowledge (e.g., on HXTs) to extract the lengthscales on which gradients emerged as a function of the external glucose concentration. Providing with a quantitative, detailed, description of these landscapes is the main scientific contribution of our work. Methodologically, our contribution is probably broader, since we describe a tool for researchers interested in the emergence of metabolic gradients in multicellular systems (microbial colonies, biofilms, multicellular tissue, tumor, etc…). As suggested by the editors, we now addressed this in the discussion part of our paper.

Reviewer #2:

Marinkovic et al. address single-cell response to glucose gradients within structured communities. They present an intriguing method to look at metabolic landscapes in a microbial community. However, I do believe the novelty of the results need to be clarified significantly better, and the conclusions drawn must be discussed more critically. Alternatively, the work may be more suitable for a methods focused publication, as using the gene expression of the transporters to gauge the extracellular concentration of glucose within a colony is an interesting tool.

We thank the reviewer for his careful review. Our goal was both to provide with a simple method to study the emergence of gradients within multicellular systems (in contract with classic studies that tend to study isolated single cells) and to obtain a quantitative description of an extended monolayer of yeast cells. As such, our analysis does not bring novel mechanistic understanding of the biology of yeast, but rather focus on bringing quantitative data that could be used to model yeast colonies.

Major concerns:A) The glucose gradients emerge in the end artificially, due to the design of the microfluidic chamber and due to the fact the yeast's consume glucose, and not due to community action. The method of the author's pictures of how the cells respond to different glucose availability as a function of glucose concentration in different areas of the closed channels. But the effects are the same if one cultivates the cells in batch cultures, and tunes glucose availability in this way. One does hence not learn anything about yeast communities, only, that yeast in batch culture and in a microfluidic chamber respond similarly to glucose depletion. Hence, is there anything new in the study, other than nicely illustrating the well-established fact that yeast activates different glucose transporter at different glucose concentration?

We disagree. The gradients do not appear artificially but because of the yeast cells. The design of the microfluidic chamber does not influence the gradients at all.If there were no cells, no gradients would be observed. We would argue that the gradients emerge precisely because there is community action. Cells at different positions consume nutrients at different rates and this is the main driving force of nutrient availability within a colony. Community structure (density, spatial structure, metabolic and genetic heterogeneity, etc.) will determine how, where and when different emergent properties arise. To put it differently, the spatial description of a yeast community cannot be learnt from single cell /batch analysis. Therefore, we do learn (quantitatively) something; the spatial pattern that emerges from single cell programs when cells are together, shaping a metabolic environment at a much larger scale than their own size.

Note that the gradient of glucose is not known inside the cell monolayer (and this is not mentioning the other nutrients) and cannot be inferred from existing models. Methods set aside, the scientific novelty of our study is that we now know quantitatively where and when different glucose transporters are expressed in space and time in a community context. It is a good fundament for further studies and models which could have hardly been extrapolated from batch culture glucose transporter expression data.

B) Batch cultures are used to tune gene expression sensors. But, in batch culture, glucose is constantly and rapidly depleted, while gene expression, needs some time to adjust. At any given time, in batch culture, the glucose concentration in the media, and the gene expression programme, are hence asynchronous (i.e. gene expression lags behind the glucose concentration). Chemostat experiments would be needed, to have a glucose concentration that corresponds to the gene expression programme at a particular point in time.

We are not sure to understand this comment. Our goal is NOT to study a particular gene in a chemostat condition. On the contrary, our method embraces the complexity of a dynamically growing cell monolayer and propose to observe and analyze its emergence and maintenance in terms of landscapes of growth and gene expression.

If this comment is directed to the way we quantified HXTs level of expression in batch culture, we do agree that the most elegant way to do this would be by using a chemostat, or even better, a microfluidic device that can trap single cells subjected to perfusion of media with a constant nutrient concentration. However, we made sure to measure glucose depletion in batch culture and adjusted our experiments accordingly (number of inoculated cells and transfer to a fresh medium) so that cells do not experience a significant drop in glucose concentration. We show in Figure 6—figure supplement 1 the concentration of glucose in supernatant right before flow cytometry measurements, and it was more than 90% of initial glucose concentration.

Method:Figure 1: I have not found the mention of the exact number of dead-end chambers within one microfluidic chip (I'm guessing 16 from the figures?). However, looking at the n in different experiments (e.g. Figure 1, n=12, n>5 and n=9) I am wondering in how many separate experiments these were done, and how much of the collected data was discarded and for which reasons?

We used different microfluidic chips but most of the experiments presented here contain 5 – 10 chamber in parallel. The number of experiments is the sum of all “growing monolayers” for a given condition, observed in several, separated experiments (2-3). Sometimes it can happen that either no or not enough cells are loaded in one or two chambers and therefore they remain empty or take a long time to fill up compared to other well-loaded chambers. Or, it can happen that during the growth some chambers get clogged by cells and then we do not take these chambers into account for analysis. In the end, we took all growing monolayers into account and did not discard any data in the analysis we provide.

Was it excluded that nutrient concentrations from the previous chamber affected the later ones? Especially in the stages when the chambers were already filling up/cells were being carried away?

Yes. The dimensions of the feeding channels and the flow rate (5μL/min) ensures that the concentration seen by the cells in different chambers are independent. In line with this consideration, we did not observe any decays of the growth rate of cells or of the speed of the monolayer front, nor any drift in the gene expression landscapes from one chambers to the next.

Results and Conclusion:Subsection “Cellular metabolic activity creates gene expression landscapes”: "Both patterns demonstrate the formation and maintenance of a glucose gradient that emerges from cellular metabolic activity." The pattern of the gene expression demonstrates that the cells adapt to the glucose gradient that emerges through supply of fresh media from only one side. "Formation and maintenance by cellular metabolic activity" in the sense that, yes, the cells close to the opening do import the glucose they require seems a weak point to make. In other words, there is evidence for the contrary – there seems no different action of cells in the community as in the batch culture, when it comes to the expression of the transporters.

We are not sure we understand the point made by the reviewer. We do not intend to show that there are “surprising” differences between what is known of yeast biology and what occurs in the monolayers. Our point, is to say that our (quantitative) knowledge of single cell (obtained e.g. in batch culture) is not enough to infer quantitatively the landscape of gene expression observed. On the contrary, we USE the landscape of gene expression to infer the glucose gradients.

Discussion paragraph four: What is the dependence/independence of the different gene expression looked at here? Can we talk about synchronicity between networks if the studied genes are known to dependent on the same external factor?

Again, this is a difference of perspective. Given that the global microenvironment is not known, it is not obvious that the different genes that we studied display a high level of coordination in their spatial patterns. For example, you could imagine that the ethanol, pH, metabolite by products, or more simply the varying dilution rate of cells due to their variable growth rate, may affect the gene expression levels. We just show that this is the case in an experimental context that is relevant for multicellular systems.

Figure 6: Landscapes of gene expression is not shown at all glucose concentrations for all genes, why?

We did not show the landscapes that were flat or non-informative. The point was to show that the landscapes were spatially correlated and adopt a structure with two main regions displaying different metabolic activity.

Reviewer #3:

This manuscript details the results of a study using a microfluidic system to study growth and gene regulation in monolayers of yeast cells. This system is presented as a tractable model for studying pattern formation in microbial colonies, via spatial and temporal measurement of cell division rates and measurement of gene expression via fluorescently tagged reporters.The authors use this experimental setup, in combination with GFP tagged version of hexose transporters, a glucose sensitive transcription factor (MIG1), and other reporters of metabolic state (PDC1, SHD2) to study spatial patterning as a function of glucose availability.Unsurprisingly, they find that patterns of expression of hexose transporters with different affinities are spatially arranged in a manner consist with a gradient of glucose availability with highest amounts of glucose available near the "source" (i.e. periphery of the artificial colony), and lowest levels near the "sink" (i.e. center of the artificial colony). Perhaps more interestingly, their interpretation of their data suggest a relatively sharp transition in the colony between zones of growth and slow/no-growth, which they ascribe to zones of fermentive and aerobic metabolism.

We understand that the reviewer found our analysis “unsurprising”, but it was not our intention to show a surprising effect (which could have been seen in a different context, for example by changing the frequency and concentration of nutrients supply or type, by perturbing gene networks or by controlling one or more additional nutrients, to name a few. Some of which are currently a subject of investigation in the lab using this microfluidic system). Here our goal was to propose a quantitative description of these landscapes. This is not only about observing two populations. It is about observing the continuum of populations and the emergence of a spatially stable structure which results from growth landscape, metabolic landscape and gene expression landscape. Methodologically, we propose a novel method that can be used to study these landscapes dynamically, which no other systems proposed so far.

While I appreciate the elegance of the microfluidic colony model that is presented, the key findings are rather modest -- i.e. there are gradients of nutrient availability in microbial colonies. This doesn't really have the impact I expect from eLife papers.

This is not the key finding of the article. It was expected that we observed gradients, but we did not know their spatial extent and their stability and maintenance. Consider, that it is not possible (to date) to propose a model that would estimate this gradient for any metabolite. This is due to our limited quantitative data that describe multicellular systems, and our limited knowledge about multicellular effects (potential synergistic or antagonistic effects, interactions with other nutrients, effects of secreted metabolites etc…). Here, we do propose quantitative measurements of these landscapes, providing in return real data to fine tune computational models. We also are able to use this to infer the shape of the glucose gradients and estimate the number of cells that can grow in such conditions. Additionally, we show that the gradients are very different depending on the amino acid concentration (Figure 2—figure supplement 1), indicating that the global microenvironment, which is also composed of complex gradients needs to be taken into account to properly model the dynamics of cellular assemblies. It is one thing to assume that there are gradients. It is not the same to propose a robust experiment that allows to quantitatively study them. This is even more important if we want to get closer to the context that occurs in natural, non-laboratory, environments where cells often find themselves in everchanging environments composed of many interacting cells of same or different type.

Below I detail a number of concerns/comments about the study, the analyses, and whether the authors have adequate related their work to the larger literature in this area.Detailed comments:1) Several other recent papers (Hornung et al., 2018,; Wilmoth et al., 2018) have used microfluidic setups to study microbial colonies. The Hornung et al. paper is the most similar to the current experimental design.

This article specifically deals with “small” chambers. There are other articles that try to model colonies, the originality of our system is that we are able to let the cells create the gradients themselves through metabolic activity over almost 1000 fold concentration range of nutrients spanning from very high concentrations that are commonly used in laboratory and that appear in nature to complete lack of nutrients. And additionally, we can track on global and single cell level the activity of cells in space and time. It is thus different and applicable to a wider range of contexts and questions. In the revised manuscript, we changed the Introduction / Discussion to mention other articles that started to address the problem of spatial organization of multicellular assemblies in microfluidics even there are only a few to our knowledge.

2) If goal was to measure the rate of glucose uptake and availability it seems like there may be other more direct approaches than inferring this from hexose transporter expression, such as the use of 2-NBDG, a non-metabolizable, fluorescent glucose analog (Roy et al. 2015, doi: 10.1371/journal.pone.0121985).

We did consider the use of 2-NBDG, however it is not as straightforward as proposed by the reviewer. As a non-metabolizable glucose analog, it is likely to be progressively absorbed by cells until they cannot absorb it anymore, leaving it to freely diffuse further in the colony. Here, glucose (and other metabolizable nutrients) show a decrease of concentration precisely because they are used and converted in biomass. Using an analog to infer the glucose concentration would likely not work.

3) The following paper, while not at the fine spatial scale provided by the microfluidics, presents a more complete and compelling view of metabolic differentiation in yeast colonies: Maršíková et al., 2017.

The interesting differentiation patterns shown in this paper and in other papers published by the same group have one shortcoming, which is that they are observed in colonies grown on glycerol or glycerol/ethanol supplemented agar plates as the major carbon source. As they are not fermentable, it is likely that the insight in this major component of yeast metabolism is limited. This method also has limited spatiotemporal scale as noted by the reviewer and inability to dynamically change environments or provide constant environments. The advantage is that it is possible to perform analysis like RNAseq, which is nicely demonstrated in this paper. In our experiments we use media that has precisely quantified components (yeast nitrogen base, complete supplement mixture of amino acids and additional carbon source at precise concentration) as opposed to media based on yeast extract (basically yeast growing on yeast, which nota bene, would be an interesting possibility to study the effects of cannibalization of dead yeast in our microfluidic colony system). Also, they do not systematically quantify the landscape of gene expression, but they showed evidence that there are different metabolic areas in a colony. We did mention this study in our revised article.

4) The spatial analyses focus primarily on peaks of maximum expression, but the data is potentially much richer and more interesting if the authors consider not only the global maxima but more complex spatial patterns. For example, in Figure 4B MIG1 nuclear localization appears to be multimodal. What is going on here?

We do agree and this is the main point that we make by quantifying the full landscape of expression of many genes in similar conditions that emerges from the metabolic activity of cells. With that regards, the conclusions may not be surprising but they take place in a physiological setting. Regarding MIG1, we do not observe a multimodal localization, but a sharp transition from a localized to a cytoplasmic state. This is used as an indicator of a shift in glucose concentration.

5) There is very little effort to contextualize this work in the larger body of studies of yeast colony structure, physiology, or regulation of glucose responsive TFs. Illustrative of this, there are only three citations total in the entire discussion. There is only a single reference to the regulation of hexose transporters in the entire manuscript. Specific examples:

We agree that more contextualization was necessary so we expanded our Introduction and Discussion part, as well as provided additional citations, including all the ones proposed by the reviewer.

- Subsection “Cellular metabolic activity creates gene expression landscapes”: " HXT1 is a low-affinity glucose transporter mainly expressed under high-glucose conditions, while HXT7 is a high-affinity glucose transporter expressed under low-glucose conditions only" Citations?

See above

- Subsection “Gene expression landscapes depend on the glucose source concentration”: "As HXT1 is mainly expressed under high-glucose conditions (> 1% w/vol glucose) in batch culture…" Citation?

See above

- Subsection “Gene expression landscapes of other genes and transcription factor activity confirm the inferred glucose gradients”: "MIG1 is a key transcription factor involved in glucose repression that localizes to the nucleus in the presence of glucose, to repress genes that participate in parallel carbon metabolic pathways (e.g., galactose)." Citation?

See above

- and in the same subsection: "…we examined the expression of PDC1 and SDH2, which are overexpressed in fermenting and respiring cells, respectively" Citation?

See above

[Editors' note: the author responses to the re-review follow.]

We really appreciate the efforts of re-writing this manuscript as a "resource" paper. We feel that the paper has been much improved, but there are some remaining issues that need to be addressed through textual changes before acceptance, as outlined below:1) We ask that the authors make it clear to the reader that what they call a "community action" does not imply that cells are responding differently in the community than they would as single cells in planktonic growth. The observed effects (gradients and response to these gradients) are merely the consequence of a high concentration of more or less static cells. So, perhaps calling it "population effect" would be more appropriate?

We did not intend to imply a “community” effect (with the meaning proposed by reviewer 2) and have exchanged all relevant instances of “community” with the word “population” as you suggested. Notably, the cells themselves are not static, *i.e.* they continuously grow and therefore passively pushed to be washed away from the channel. However, a steady state gradient is achieved so that cells, transiently occupying a geographic location along the gradient axes, exhibit a steady state response, mirroring the established stable gradient.

2) We would appreciate if you could insert a succinct discussion of the difference between a batch culture and this steady-state (where in principle, the cells could be better adapted as they are not always "running behind" in their response to a variable environment, as they would be in batch). (see reviewer 2's comments below for more details)

i) In standard batch culture, as cells exhaust the media, their adaptation time, limited by sensing, transcription and translation (on the order of 30 minutes; response time referred to by reviewer #2), may lag behind the decrease of glucose in the media. In contrast, in our 2D colony device, once steady-state gradient is settled, cells’ residence time within a given range of glucose that corresponds to stable Hxt expression is significantly longer, assuring that even though cells are continuously pushed away, they spend sufficient time within a given concentration to equilibrate their response to the gradient and therefore faithfully report the glucose concentration in their environment. As example, at 1% glucose the mean velocity at Hxt7 expression peak, spanning over >200μm (Figure 3A, varies between 5-15μm/hour (Figure 2F), suggesting cells’ residence time of >10 hours, pointing to the advantage of working in this setup.

ii) In our batch culture experiments (modified from a previously published protocol reported by Youk and van Oudenaarden, 2009), we intently used very diluted cultures to assure that within the experiment span cells both adapt to the nutrient condition and do not consume sufficiently to change significantly the glucose concentration. Thus, we deem the chemostat experiment is not useful.

We addressed point (i) by inserting the paragraph above to the Discussion section, and point (ii) in a succinct manner in the Materials and methods section of our revised manuscript.

3) We again ask to include more references and to not overstate the advantage of the presented device (see reviewer 3's comment below).

We apologize to the reviewer for the confusion about the references that we actually added in the revised manuscript but not pointing to the correct line numbers. The lines were referring to the lines in the original manuscript that the reviewer pointed out. In the current revised manuscript they can be found in subsections “Cellular metabolic activity creates gene expression landscapes” and “Gene expression landscapes of other genes and transcription factor activity confirm the inferred glucose gradients”.

Indeed, our method compliments other approaches cited in the manuscript and expands researchers’ arsenal for quantitative observation of 2D colony growth. We addressed this in the revised manuscript, toned down the advantages of our device and discussed its limitations in view of other published devices more clearly (see Introduction section).

4) Please provide more details about the reproducibility/variability and statistics of the results and analyses.

Each result is represented with standard deviation as error bars as descriptors of how disperse measurements are. When we mentioned the reproducibility of our experiments, it is based on the fact that error bars are small compared to the range of variation of the mean fluorescence peak location when the control parameter (here the external glucose concentration) is varied. Given the significant in-between concentration differences as compared to the error bars we see no added value in computing further statistics such as p-values. All figure captions contain the number of samples and the description of the error bars displayed. In addition, the complete dataset used in this study is available on an online Zenodo archive.